# Team-Fictitious Play for Reaching Team-Nash Equilibrium in Multi-team Games

**Ahmed Said Donmez**
Bilkent University
`said.donmez@bilkent.edu.tr`

**Yuksel Arslantas**
Bilkent University
`yuksel.arslantas@bilkent.edu.tr`

**Muhammed O. Sayin**
Bilkent University
`sayin@ee.bilkent.edu.tr`

## Abstract

Multi-team games, prevalent in robotics and resource management, involve team members striving for a joint best response against other teams. Team-Nash equilibrium (TNE) predicts the outcomes of such coordinated interactions. However, can teams of self-interested agents reach TNE? We introduce Team-Fictitious Play (Team-FP), a new variant of fictitious play where agents respond to the last actions of team members and the beliefs formed about other teams with some inertia in action updates. This design is essential in team coordination beyond the classical fictitious play dynamics. We focus on zero-sum potential team games (ZSPTGs) where teams can interact pairwise while the team members do not necessarily have identical payoffs. We show that Team-FP reaches near TNE in ZSPTGs with a quantifiable error bound. We extend Team-FP dynamics to multi-team Markov games for model-based and model-free cases. The convergence analysis tackles the challenge of non-stationarity induced by evolving opponent strategies based on the optimal coupling lemma and stochastic differential inclusion approximation methods. Our work strengthens the foundation for using TNE to predict the behavior of decentralized teams and offers a practical rule for team learning in multi-team environments. We provide extensive simulations of Team-FP dynamics and compare its performance with other widely studied dynamics such as smooth fictitious play and multiplicative weights update. We further explore how different parameters impact the speed of convergence.

## 1 Introduction

Multi-team games are increasingly common, e.g., in robotics and resource management [Silva and Chaimowicz, 2017, Vinyals et al., 2019, Jaderberg et al., 2019]. Unlike non-cooperative multi-agent settings, team members strive for a joint best response against other teams as if the entire team is a single decision-maker. Team-Nash equilibrium (TNE), where team members coordinate in the best team response against other teams, can capture this to predict the outcome of coordinated team interactions [Farina et al., 2018, Zhang et al., 2021]. Game-theoretical equilibrium is often justified by its emergence from non-equilibrium adaptation of self-interested learners (e.g., see [Fudenberg and Levine, 2009]). However, the question of whether the teams of self-interested agents can reach TNE in multi-team games remains largely unexplored. This paper investigates this very question.

For example, TNE generally can arise if the team members can learn to correlate their actions in the best team response independent of the opponent. However, the widely studied fictitious play (FP) dynamics do not necessarily reach the best team outcome even when there are no opponents, e.g., in

38th Conference on Neural Information Processing Systems (NeurIPS 2024).

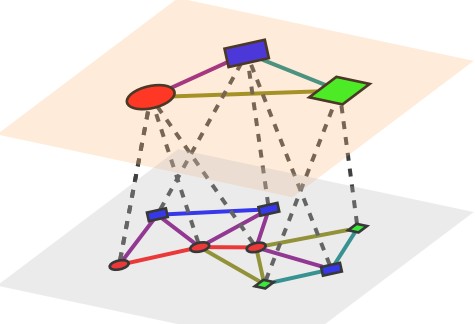

Figure 1: An illustration of networked interconnections agents from different teams. Nodes in bottom and top layers refer, resp., to team members and teams. Undirected edges represent the impact of actions on the payoff functions. We use different colors and shapes to represent agents from the same teams, and they are connected via dashed edges.

potential games. We present a slight adjustment of FP, called *Team-FP*, provably reaching TNE in multi-team competition even with networked interconnections, where agents' payoffs depend only on the neighbors' actions. Similar to the FP, here, agents respond greedily to the beliefs formed about the opponent teams' joint play based on past observations. Different from the FP, Team-FP incorporates the key features: $(i)$ response to the last actions of team members, and $(ii)$ inertia in action updates. These features, inspired by log-linear learning dynamics [Blume, 1993], play a crucial role in driving team coordination towards TNE. Notably, Team-FP reduces to the smoothed FP [Fudenberg and Kreps, 1993] (or log-linear learning) when each team has a single agent (or there is a single team).

Multi-team competition spans diverse domains. For example, robotics, resource management, online gaming, and financial markets [Kitano et al., 1997, Cardenas et al., 2009, Silva and Chaimowicz, 2017, Vinyals et al., 2019, Jaderberg et al., 2019] involve multi-team competition. To model such interactions, we consider multiple teams with possibly different number of team members. These team members have networked interconnections, as depicted in Figure 1. We focus on multi-team *zero-sum potential team games* where teams have *pairwise interactions* (ZSPTGs). For any opponent team strategy, team members effectively play an underlying potential game, as in distributed optimization applications, e.g., see [Arslan et al., 2007, Xu et al., 2012, Zheng et al., 2014]. Notably, ZSPTGs reduce to zero-sum polymatrix games [Cai et al., 2016] if each team has a single agent, and to potential games [Monderer and Shapley, 1996a] if there is a single team. Additionally, widely studied two-team zero-sum games, e.g., see [Farina et al., 2018, Zhang et al., 2021, Carminati et al., 2022, Kalogiannis et al., 2022], are a special case of ZSPTGs.

We show that the Team-FP dynamics reach near TNE in ZSPTGs. This means that the empirical average of team actions converge to the near best response each team can take against the average actions of other teams. We quantify the approximation error, showing it decreases with the level of exploration in the agents' responses. Similar to the FP dynamics, Team-FP is also *rational*: teams can learn (near) optimal strategies if opponent teams play stationary strategies. These results strengthen the applicability of TNE for predicting team behavior in multi-team competition and provide a practical rule for teams of self-interested agents to learn coordination in multi-team settings.

A key challenge in our analysis is handling the non-stationary nature of learning, as opponent teams' strategies change over time. We address this by leveraging the optimal coupling lemma (e.g. see [Levin and Peres, 2017, Chapter 4]) and stochastic differential inclusion approximation methods (e.g., see [Benaïm et al., 2005, Perkins and Leslie, 2013]) to the repeated play of games. Motivated from the recent interest in multi-agent reinforcement learning, we can extend Team-FP dynamics to finite horizon multi-team Markov games for both model-based and model-free cases. We discuss this extension and analyze its convergence numerically in Appendix C.

**Related works.** FP and its variants offer convergence guarantees in important classes of games [Fudenberg and Levine, 2009], yet not in every class of games [Hart and Mas-Colell, 2003]. For example, they reach equilibrium in potential games [Monderer and Shapley, 1996a], but not necessarily the most efficient one for the team. Log-linear learning can achieve efficient equilibrium for the team [Marden and Shamma, 2012, Tatarenko, 2017]. However, it is not clear whether such dynamics can track efficient equilibrium in dynamic environments (induced by the evolving strategies of opponent teams). Notably, Tatarenko [2018] and Donmez et al. [2024] addressed, resp., efficient learning under non-stationarity induced by the decaying exploration in agents' responses for the repeated play of potential games and non-stationarity induced by evolving stage games in Markov team problems

(also known as identical-interest Markov games). These approaches are orthogonal to our analysis to extend our results to the exact TNE convergence in repeated multi-team games or to learning in infinite horizon Markov games.

FP and its variants can reach equilibrium in two-agent zero-sum games [Hofbauer and Sandholm, 2002] yet not necessarily in multi-agent zero-sum games with more than two agents. We can transform any general-sum game to a multi-agent zero-sum game by introducing a non-effective auxiliary agent (with a single action). There have been several attempts to address zero-sum games beyond two-agent cases [Bergman and Fokin, 1998, Cai and Daskalakis, 2011, Cai et al., 2016]. For example, Ewerhart and Valkanova [2020] addressed the convergence of continuous and discrete-time FP in zero-sum network games, where each agent plays multiple two-agent games with separate actions, and the overall utilities sum to zero. Notably, Cai et al. [2016] introduced zero-sum polymatrix games where agents have network separable pairwise interactions with applications in security. Recently, FP has been shown to reach Nash equilibrium in zero-sum polymatrix games [Park et al., 2023]. Following the same trend, we focus on learning in ZSPTGs extending two-team zero-sum games to multi-team games with pairwise team interactions. However, we highlight that two-team zero-sum or ZSPTGs are not necessarily multi-agent zero-sum polymatrix games (as the agent payoffs do not necessarily sum to zero) and the classical FP dynamics do not necessarily converge TNE or Nash equilibrium in such multi-team games.

Recent studies on two-team zero-sum games and adversarial team games (e.g., see [Celli and Gatti, 2018, Farina et al., 2018, Zhang et al., 2022, Carminati et al., 2022]) have primarily focused on the efficient computation of team-minimax equilibrium. In particular, Celli and Gatti [2018] examines the efficiency of different communication types, highlighting the promising results of *ex ante* communication, referring to pre-play communication among team members. Consequently, the studies of Farina et al. [2018], Zhang et al. [2022], Carminati et al. [2022] often model teams as a single agent with imperfect recall, incorporating ex ante communication within the team. Notably, Farina et al. [2018] introduced the Fictitious Team Play (FTP) algorithm for extensive-form two-team zero-sum games with imperfect information, where team members communicate ex ante. In this approach, Farina et al. [2018] used fictitious play (FP) on a simplified version of the original game, embedded within the game tree. This method essentially applied FTP to the original adversarial team game. To find the best response, they used mixed-integer linear programming. Team-FP differs from such approaches by letting agents *learn* to team up while following their self-interest based on simple behavioral rules, as in the log-linear learning and FP dynamics.

The rest of the paper is organized as follows. We describe ZSPTGs in §2. We present the (independent) Team-FP dynamics in §3. We provide analytical and numerical results, resp., in §4 and §5. We conclude the paper with some remarks in §6. Appendices include the proofs of the technical results, some further numerical experiments and the extension to multi-team Markov games.

*Notation:* Given a finite set $A$, we let $\Delta(A)$ denote the probability simplex over $A$. We let $f(\mu) = \mathrm{E}_{a \sim \mu}[f(a)]$ for any probability distribution $\mu \in \Delta(A)$ and any functional $f : A \to \mathbb{R}$. Furthermore, we define the smoothed best response to any functional $q : A \to \mathbb{R}$ by

$$\mathrm{br}_\tau(q)(a) = \frac{\exp\left(q(a)/\tau\right)}{\displaystyle\sum_{\widetilde{a} \in A} \exp\left(q(\widetilde{a})/\tau\right)} \quad \forall a \in A \quad \Leftrightarrow \quad \mathrm{br}_\tau(q) = \underset{\mu \in \Delta(A)}{\mathrm{argmax}}\{q(\mu) + \tau\mathcal{H}(\mu)\}, \quad (1)$$

for some temperature parameter $\tau > 0$, where $\mathcal{H}(\mu) := \mathrm{E}_{a \sim \mu}[-\log(\mu(a))]$ for all $\mu \in \Delta(A)$ is the entropy regularization.

## 2 Game Formulation

Consider a *multi-team game*, characterized by the tuple $\langle \mathcal{T}, (A^i, u^i)_{i \in \mathcal{I}} \rangle$, where $\mathcal{T}$ and $\mathcal{I}$ denote, resp., the teams' and agents' index sets, and $A^i$ and $u^i : A \to \mathbb{R}$ (with $A := \prod_{j \in \mathcal{I}} A^j$) denote, resp., the agent $i$'s finite action set and payoff function. Agents take their actions to maximize their payoff functions.

**Definition 2.1** (Zero-sum Potential Team Game). Let $\mathcal{I}^m$ denote the index set of agents in team $m$ and $\underline{A}^m := \prod_{i \in \mathcal{I}^m} A^i$ denote the set of joint actions for team $m$. We say that a multi-team game is *zero-sum potential team game* (ZSPTG) if for each team $m \in \mathcal{T}$, there exists a potential function

$\phi^m : A \to \mathbb{R}$ such that

$$u^i(\hat{a}^i, a^{-i}, \underline{a}^{-m}) - u^i(a) = \phi^m(\hat{a}^i, a^{-i}, \underline{a}^{-m}) - \phi^m(a), \tag{2}$$

for all $(\hat{a}^i, a) \in A^i \times A$ and $i \in \mathcal{I}^m$, where $a^{-i} := \{a^j\}_{j \in \mathcal{I}^m \setminus \{i\}}$ are the actions of other team members, $\underline{a}^{-m} := \{\underline{a}^\ell\}_{\ell \neq m}$ are the action profiles of other teams, where $\underline{a}^\ell \in \underline{A}^\ell$ is team $\ell$'s action profile. The potential functions sum to zero, i.e., we have

$$\sum_{m \in \mathcal{T}} \phi^m(a) = 0 \quad \forall a \in A. \tag{3}$$

Furthermore, the actions have network separable interactions across teams such that we can separate the potential functions and correspondingly payoff functions as

$$\phi^m \equiv \sum_{\ell \neq m} \phi^{m\ell} \quad \text{and} \quad u^i \equiv \sum_{\ell \neq m} u^{i\ell} \quad \forall i \in \mathcal{I}^m, \tag{4}$$

for some $\phi^{m\ell} : \underline{A}^m \times \underline{A}^\ell \to \mathbb{R}$ and $u^{i\ell} : \underline{A}^m \times \underline{A}^\ell \to \mathbb{R}$.

The following example generalizes the potential game formulation for distributed optimization (e.g., see [Arslan et al., 2007, Xu et al., 2012, Zheng et al., 2014]) to two-team zero-sum potential games.

*Example* 2.2. Consider two teams of agents interacting over a network. We can represent their interactions via a graph $G = (V, E)$, where the set of vertices $V$ refers to the agents and the set of (undirected) edges refers to their interactions. Let agent $i$ from team $m$ receive a local payoff $r^i : A^i \times \prod_{j:(i,j) \in E} A^j \to \mathbb{R}$ depending on the actions of the neighboring agents only. Agent $i$ adds the neighboring team members' local payoffs whereas subtracts the other neighbors' local payoffs in his/her total payoff. In other words, his/her total payoff is given by

$$u^i \equiv \sum_{j:(i,j) \in E} \mathbb{I}_{\{j \in \mathcal{I}^m\}} r^j - \sum_{j:(i,j) \in E} \mathbb{I}_{\{j \notin \mathcal{I}^m\}} r^j. \tag{5}$$

This yields that the team $m$ has the potential function

$$\phi^m \equiv \sum_{j \in \mathcal{I}^m} r^j - \sum_{j \notin \mathcal{I}^m} r^j, \tag{6}$$

and therefore, these potential functions sum to zero. However, the potential function is not generally the sum of the payoffs in potential games.

*Remark* 2.3 (General-sum ZSPTGs). In ZSPTGs, the underlying game can be a *general-sum* game although the team-potentials sum to zero, as described in (3). For example, consider two competing teams whose team members have identical payoffs corresponding to their team potentials. If the teams have different number of members, then the agents' payoffs do not sum to zero or a constant while the team potentials do so.

In the following, we introduce TNE, generalizing team-minimax equilibrium for two-team zero-sum games, e.g., see [Von Stengel and Koller, 1997], to multi-team games. Particularly, at TNE, no team has an incentive to change their team strategy.

**Definition 2.4** (Team-Nash Gap). Given the strategy profile of teams $\{\pi^m \in \Delta(\underline{A}^m)\}_{m \in \mathcal{T}}$, we define the *team-Nash gap* for team $m$ as

$$\text{TNG}^m(\pi) := \max_{\widetilde{\pi} \in \Delta(\underline{A}^m)} \left\{ \phi^m(\widetilde{\pi}, \pi^{-m}) \right\} - \phi^m(\pi), \tag{7}$$

and $\text{TNG}(\pi) := \sum_{m \in \mathcal{T}} \text{TNG}^m(\pi)$, where $\pi^{-m} := \{\pi^\ell\}_{\ell \neq m}$. Correspondingly, we say that the strategy profile of teams $\{\pi^m\}_{m \in \mathcal{T}}$ is $\epsilon$-*TNE* if $\text{TNG}(\pi) \leq \epsilon$.

## 3 Team-FP Dynamics

We first present the Team-FP dynamics combining the log-linear learning and fictitious play for learning in multi-team games played repeatedly, and then extend Team-FP to multi-team MGs in Appendix C.

**Algorithm 1** (Independent) Team-FP

---

**initialize:** $\{\pi_0^\ell\}_{\ell \neq m}$ and $\{a_{-1}^j\}_{j \in \mathcal{I}^m \setminus \{i\}}$ arbitrarily
**for** each stage $k = 0, 1, \ldots$ **do**
  play $a_k^i \sim \mathrm{br}_\tau(u^i(\cdot, a_{k-1}^{-i}, \pi_k^{-m}))$ or $a_k^i = a_{k-1}^i$ in a coordinated way (or independently)
  update $\pi_{k+1}^\ell = \pi_k^\ell + \alpha_k(\underline{a}_k^\ell - \pi_k^\ell)$ for all $\ell \neq m$
**end for**

---

Let $a_k^i \in A^i$ denote the agent $i$'s action at the $k$th repetition in the repeated play of the underlying ZSPTG. Correspondingly, $\underline{a}_k^m = (a_k^i)_{i \in T_m}$ denote the team $m$'s action profile. Observing the joint actions of team $m$, agents $j \notin \mathcal{I}^m$ can form a belief about the team $m$'s joint strategy. Let $\pi_k^m \in \Delta(\underline{A}^m)$ denote the belief they formed. Consider actions as pure strategy where the associated action gets played with probability 1. Then, agents $j \notin \mathcal{I}^m$ can update their beliefs about team $m$'s strategy according to

$$\pi_{k+1}^m = \pi_k^m + \alpha_k(\underline{a}_k^m - \pi_k^m) \quad \forall k = 0, 1, \ldots \tag{8}$$

such that the belief $\pi_{k+1}^m$ also corresponds to the (weighted) empirical average of the past action profiles $\{\underline{a}_0^m, \ldots, \underline{a}_k^m\}$.

Agent $i \in \mathcal{I}^m$ either takes the previous action $a_{k-1}^i$ (i.e., $a_k^i = a_{k-1}^i$), or takes the action $a_k^i \sim \mathrm{br}_\tau(u^i(\cdot, a_{k-1}^{-i}, \pi_k^{-m}))$ according to the smoothed best response (as described in (1)) to the previous actions of team members $a_{k-1}^{-i} := \{a_{k-1}^j\}_{j \in \mathcal{I}^m \setminus \{i\}}$ and the beliefs $\pi_k^{-m} := \{\pi_k^\ell\}_{\ell \neq m}$ formed about other teams. It is instructive to note that the definition of potential function $\phi^m(\cdot)$, as described in (2), yields that

$$\mathrm{br}_\tau\big(u^i(\cdot, a_{k-1}^{-i}, \pi_k^{-m})\big) \equiv \mathrm{br}_\tau\big(\phi^m(\cdot, a_{k-1}^{-i}, \pi_k^{-m})\big) \quad \forall i \in \mathcal{I}^m. \tag{9}$$

We introduce Team-FP and independent Team-FP dynamics depending on how agents update their actions. In the former, a single agent can get chosen randomly, as in the classical log-linear learning. In the latter, each agent can update his/her action with probability $\delta \in (0, 1)$ independent of others, as in the independent log-linear learning. The latter addresses the coordination burden in the update of actions within teams. Algorithm 1 provides descriptions of these dynamics for the typical agent $i$ from team $m$.

*Remark* 3.1 (Scalability). Agents can have networked interactions such that their payoff functions depend only on the actions of certain agents such as one/two-hop neighbors. For such cases, agents can form beliefs about these neighbors' strategies only, as if these neighbors play according to some stationary strategy. For example, assume that the payoff of agent $i \notin \mathcal{I}^m$ (outside team $m$) depends *only* on the actions of team-$m$ members from some neighborhood $\mathcal{N}^i$, i.e., $\{j : j \in \mathcal{N}^i \cap \mathcal{I}^m\}$. Agent $i$ can form a belief about these agents' strategies according to

$$\pi_{k+1}^{im} = \pi_k^{im} + \alpha_k(\underline{a}_k^{im} - \pi_k^{im}) \quad \forall k = 0, 1, \ldots \tag{10}$$

where $\underline{a}_k^{im} = \{a_k^j\}_{j \in \mathcal{N}^i \cap \mathcal{I}^m}$. Then, the linearity of the update rule yields that the local empirical average $\pi_k^{im}$ corresponds to the marginalization of $\pi_k^m$ such that

$$\pi_k^{im}(\{a^j\}_{j \in \mathcal{N}^i \cap \mathcal{I}^m}) = \sum_{\{a^j\}_{j \in \mathcal{I}^m \setminus \mathcal{N}^i}} \pi_k^m(\{a^j\}_{j \in \mathcal{I}^m}) \quad \forall k. \tag{11}$$

Therefore, local observations (within neighborhoods) would still be sufficient to follow Algorithm 1. We demonstrate the scalability of Team-FP by a large-scale experiment in Appendix D, Figure 7.

We focus on the homogeneous cases where agents $i \notin \mathcal{I}^m$ have a common belief about team $m$'s strategy. Homogeneous beliefs are possible if the agents have common initial beliefs and step sizes.

We moved the extension of (Independent) Team-FP to multi-team Markov games and its numerical analysis to Appendix C.

## 4 Convergence Results

Team-FP and Independent Team-FP dynamics reduce, resp., to the classical log-linear learning and independent log-linear learning dynamics if there is only one team. These log-linear learning

dynamics are known to reach team-optimal solution in potential games. For example, consider an $n$-agent potential game $\langle (A^i, u^i)_{i \in [n]} \rangle$ with the potential function $\phi(\cdot)$. In both dynamics, the action profiles played form irreducible and aperiodic Markov chains. Let $\widehat{\mu}$ and $\breve{\mu}$ denote the unique stationary distributions, resp., for the classical and independent versions. For the former, we have $\widehat{\mu} = \mathrm{br}_\tau(\phi(\cdot))$ [Marden and Shamma, 2012, Section 3] and (1) yields that

$$0 \leq \max_a \{\phi(a)\} - \phi(\widehat{\mu}) \leq \tau \log |A|. \tag{12}$$

On the other hand, the smaller $\delta > 0$ implies closer stationary distributions in the classical and independent versions. Particularly, we have

$$\|\widehat{\mu} - \breve{\mu}\|_1 \leq \Lambda(\delta, \epsilon_\phi), \tag{13}$$

for some function $\Lambda(\cdot)$ decaying to zero as $\delta \to 0^+$ for any $\epsilon_\phi > 0$, and $0 < \epsilon_\phi \leq \mathrm{br}_\tau(\phi(\cdot, a^{-i}))$ for any $a^{-i}$ and $i$ is a lower bound on local actions get played in the smoothed best response [Donmez et al., 2024, Lemma 5.6].

Team-FP dynamics have similar convergence guarantees for multi-team games under the following assumption on the step sizes used.

**Assumption 4.1.** The step size $\alpha_k \in [0, 1]$ satisfies the stochastic approximation conditions: $\alpha_k \to 0$ as $k \to \infty$, $\sum_{k=0}^\infty \alpha_k = \infty$, and $\sum_{k=0}^\infty \alpha_k^2 < \infty$. Additionally, we have $\lim_{k \to \infty} \alpha_k / \alpha_{k+1} = 1$, and $\alpha_k - \alpha_{k+1} \geq \alpha_k \alpha_{k+1}$.

The last condition in Assumption 4.1 ensures that recent observations have comparable weight in the beliefs formed. The classical choice $\alpha_k = 1/(k+1)$, leading to the empirical averages of the actions played, satisfies Assumption 4.1.

The following theorem shows the convergence of Algorithm 1 to near TNE almost surely with the approximation levels (similar to (12) and (13)) inherited from the (independent) log-linear learning dynamics.

**Theorem 4.2.** *Given a ZSPTG characterized by $\langle \mathcal{T}, (A^i, u^i)_{i \in \mathcal{I}} \rangle$, let every agent follow either Team-FP or Independent Team-FP, as described in Algorithm 1. If Assumption 4.1 holds, then the team-Nash gap for $\pi_k := (\pi_k^m)_{m \in \mathcal{T}}$ satisfies*

$$\limsup_{k \to \infty} \mathrm{TNG}(\pi_k) \leq \begin{cases} \tau \log |A| & \text{for Team-FP} \\ \tau \log |A| + |\mathcal{T}|^2 \overline{\phi} \cdot \Lambda(\delta, \epsilon) & \text{for Independent Team-FP} \end{cases} \tag{14}$$

*almost surely, where $\overline{\phi} := \max_{(m,l,a)} |\phi^{ml}(a)|$.*

The key challenge in our convergence analysis is the non-stationarity arising from opponent teams adapting their strategies while the team members learn to coordinate against them. We address this by constructing a *reference scenario* where the team members' beliefs about opponent strategies are *frozen* over finite-length epochs, allowing them to hypothetically "team up" under these fixed beliefs. By comparing the dynamics in the actual scenario and the reference scenario, and exploiting the averaging nature of belief formation, we can bound the approximation error between the two. The main proof concept, along with the Team-FP algorithm for a two-team scenario, is illustrated in Figure 2. This approach is similar to the one used in [Donmez et al., 2024] to handle non-stationarity in Markov team problems. However, unlike [Donmez et al., 2024], we cannot directly show the error bound decay due to the lack of a contraction property in our dynamics.

To overcome this limitation, we view Team-FP as smoothed fictitious play dynamics in zero-sum polymatrix games with an additive bounded error term. The additive error captures the fact that team members may not perfectly achieve team coordination. We then relax the problem by considering any approximation error within the formulated bounds, rather than the actual error. To address this relaxation, we leverage stochastic differential inclusion approximation methods [Benaïm et al., 2005, Perkins and Leslie, 2013]. Finally, by constructing a suitable Lyapunov function addressing arbitrary bounded errors in continuous-time smoothed best response dynamics, we establish the convergence of the discrete-time Team-FP updates.

The following corollary to the main result shows the rationality of the (independent) Team-FP dynamics.

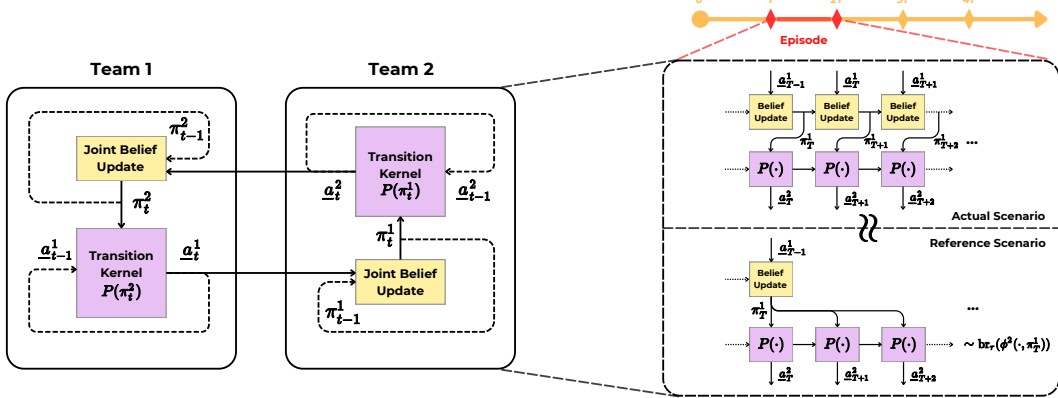

Figure 2: An illustration of Team-FP dynamics for two-team games on the left-hand side. Team actions change according to a transition kernel depending on the beliefs formed about the other teams. Dashed lines represent the time shift. On the right-hand side, we depict the key proof idea that we approximate the evolution of the team actions with a reference scenario where beliefs are stationary such that team actions form a homogeneous Markov chain whose unique stationary distribution corresponds to the best team response.

**Corollary 4.3.** *Given a ZSPTG characterized by $\langle \mathcal{T}, (A^i, u^i)_{i \in \mathcal{I}} \rangle$, let agents from team $m$ follow either Team-FP or Independent Team-FP while other teams play according to some stationary strategy $\pi^{-m}$. If Assumption 4.1 holds, then empirical average of the action profiles played by team $m$ satisfy*

$$\limsup_{k \to \infty} \mathrm{TNG}^m(\pi_k^m, \pi^{-m}) \leq \begin{cases} \tau \log |\underline{A}^m| & \text{for team-FP} \\ \tau \log |\underline{A}^m| + \overline{\phi} \cdot \Lambda(\delta, \epsilon) & \text{for Independent Team-FP} \end{cases} \tag{15}$$

*almost surely.*

Theorem 4.2 can be generalized to the case where the rewards are random with bounded support, rather straightforwardly. Therefore, the proof of Corollary 4.3 follows from Theorem 4.2 if we view the underlying game as there is a single team receiving random rewards with bounded support.

# 5 Illustrative Examples

In this section, we present various simulation results demonstrating the coordination speed of Team-FP and compare it to pure FP, no-regret algorithms, and a stationary opponent. We also observe the effect of parameters on the convergence speed. In addition, we examine the long-run behavior of team-FP in games beyond ZSPTG games, where we intuitively expect it to converge. All simulations are averaged over 10 independent trials to reduce the randomness. In all figures, the mean is plotted with a thick colorful line, while one standard deviation below and above the mean is shown with a shaded area of the same color. Also, for all simulations, temperature parameter $\tau$ is chosen to be 0.1 unless another option is mentioned. We conduct simulations for ZSPTG with two different setups: one with three teams, each consisting of three agents, and another with two teams, each consisting of four agents. Unless explicitly stated otherwise, the default setting consists of two teams, with four agents in each team. The step size is chosen to be $\alpha_k = 1/(k+1)$ for all simulations.

All the simulations are executed on a computer equipped with an Intel Xeon W7-3455 CPU and 128 GB RAM. Run-time for 10 independent trials over $10^6$ iterations vary between 1-5 hours depending on the experiment.

**Achieving Implicit Coordination in Team-FP** In this section, we compare the performance of Team-FP to the explicit coordination of team members. We also compare Team-FP to algorithms such as Multiplicative Weights Update (MWU) and Smoothed FP (SFP), and show that these algorithms fail to achieve team coordination. We also show that Team-FP achieves near-optimum policy against a stationary opponent as stated in Corollary 4.3.

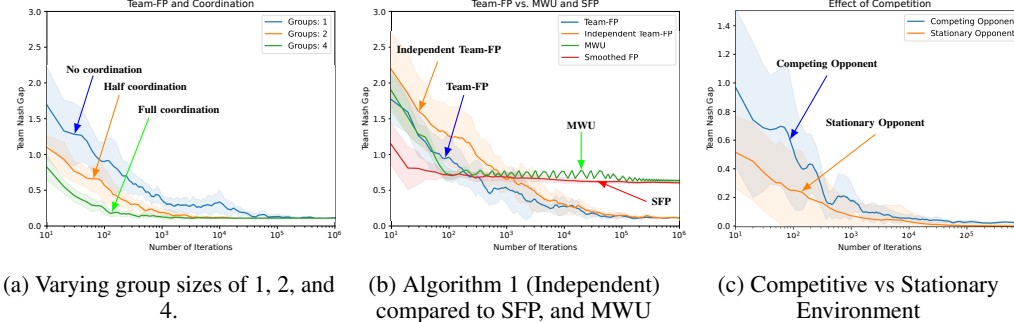

(a) Varying group sizes of 1, 2, and 4.

(b) Algorithm 1 (Independent) compared to SFP, and MWU

(c) Competitive vs Stationary Environment

Figure 3: All the above figures show the variation of TNG over time. (a) Comparison of different levels of explicit coordination for Team-FP: independent agents (group size 1), pairs of cooperating agents (group size 2), and fully coordinated teams (group size 4). (b) Performance of Team-FP and Independent Team-FP compared to MWU and SFP algorithms in a 2-team ZSPTG. (c) Convergence of Team-FP against stationary and competitive opponents in a 3-team ZSPTG.

*Impact of Team Size in Team Coordination:* In Team-FP self-interested team members act together without explicit communication in a coordinated way to reach TNE. We can measure how this independent cooperation compares to the explicit coordination of team members. For that, we propose an example game with two teams and four agents in each team. For the first scenario, four agents act separately and use Team-FP. In the second scenario, the agents form groups of two and act in a coordinated way as if they are a single agent, using Team-FP. In the final scenario, all agents in a team behave as if they were a single agent, equivalent to the standard fictitious play (FP) dynamics in a two-person zero-sum game. This case mimics the ex-ante communication scheme from [Farina et al., 2018]. The scenarios are described by group sizes. Group sizes are one, two, and four respectively for these scenarios. The simulation results are presented in Figure 3a. As expected, the explicit coordination of all members in a team converges the fastest, followed by the explicit coordination of groups of two, and finally, Team-FP converges slowly as the agents do not coordinate explicitly.

*Team-FP Compared to MWU and SFP:* The strong side of Team-FP is, even though the agents do not communicate, the average of joint actions of teams can reach TNE, unlike other algorithms. We compare the equilibrium behavior of team-FP with a well-known no-regret learning algorithm Multiplicative Weights Update (MWU), in which the average strategies converge to NE in zero-sum polymatrix games [Cai et al., 2016]. We also compare Team-FP with the usual SFP, where each agent holds beliefs about other agents and uses the smoothed best response against them. In Figure 3b, we see that both Team-FP and Independent Team-FP dynamics converge to near TNE, while the other algorithms fail to do so.

*Rationality Against Stationary Opponent*: In this part, we use 3-team setting where each team has 3 agents. We let a team using Team-FP compete against two other stationary teams and compared it with the performance of the same team in the same game when the opponents are also using Team-FP competitively. In Figure 3c, we observe that both algorithms converge, while the convergence is much faster against stationary opponents.

**Team-FP in Application** In this part, we provide an example to demonstrate that Team-FP has applications in various contexts.

*Security Game Example*: We model an airport security scenario as a two-team game between defender and attacker teams. In our example, a security chief on the defender team faces three independent intruders on the attacker team, as shown in Figure 4a. The chief can defend a gate at a cost, while intruders decide whether to attack a gate or remain idle. Intruders gain or lose payoffs based on whether they attack undefended or defended gates, and the chief's payoffs mirror these outcomes. We conducted multiple trials and presented the evolution of the average Team-Nash Gap with standard deviations on the right side of Figure 4b. From a higher level, this example shows that team-minimax equilibrium can predict the outcome of games with different uncoordinated attackers. It also justifies the algorithms developed to compute team-minimax equilibrium efficiently.

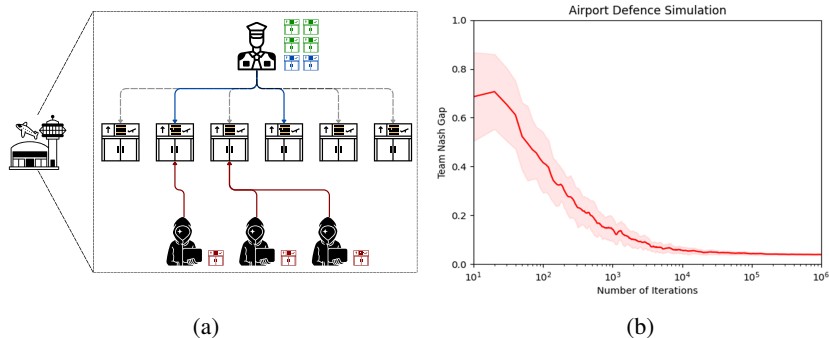

(a)                                                                (b)

Figure 4: (a) The illustration of an airport security game: a security chief guarding the six gates of an airport against three different intruders making decisions autonomously. (b) The evolution of Team Nash Gap in airport security game, showing that Team-FP dynamics reach near team-minimax equilibrium.

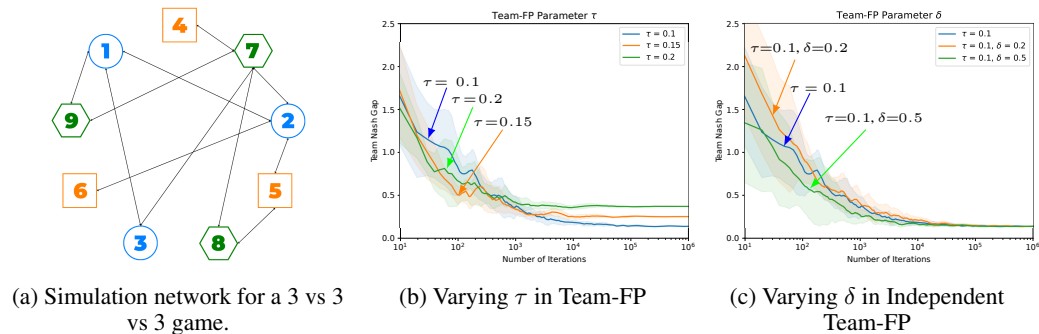

(a) Simulation network for a 3 vs 3 vs 3 game.

(b) Varying $\tau$ in Team-FP

(c) Varying $\delta$ in Independent Team-FP

Figure 5: The 3-team experiments are tested on the randomly generated network structure (a). The other figures (b), and (c), shows the variation of TNG over iterations. (a) The simulation network for a multi-team ZSPTG, in which there are 3-teams with 3 agents in each team. (b) The impact of varying temperature parameter $\tau$ (0.1, 0.15, 0.2) in Algorithm 1 on the closeness to TNE. (c) The effect of different $\delta$ values (0.2, 0.5) in (Independent) Algorithm 1 on the convergence speed with $\tau$ fixed at 0.1

**Effect of parameters in Team-FP**   In this part, we examine how Team-FP performs for different $\tau$ and $\delta$ values. Given an example ZSPTG of three teams where each team has three agents in Figure 6a, we examine the evolution of TNG in the Team-FP dynamics for different values of $\tau \in \{0.1, 0.15, 0.2\}$. We also compare the evolution of NG in the Team-FP and Independent Team-FP dynamics for $\tau = 0.1$, $\delta = 0.2$, $\delta = 0.5$. All simulation results (see Figure 5) show convergence, and we observe lower final values of NG$(\pi)$ for smaller $\tau$ as we predicted (see. Figure 5b). The Independent Team-FP requires more iterations when $\delta = 0.2$ for its convergence, while it is much faster when $\delta = 0.5$ (see. Figure 5c). This is expected as updates are much more frequent as $\delta$ increases. However, increasing $\delta$ too much may harm the coordination behavior of the team.

**Beyond ZSPTG**   In this part, we try several other games for Team-FP without proof of convergence. We expect Team-FP to converge in 2xN and potential games other than zero-sum games as FP converges in these settings. For the first case, we try a game where one team has a single agent with only two actions with random rewards, while the other team has three agents with an underlying potential function. In this case, Team-FP converges (see. Figure 6b). In another setting, we try two teams of four agents. However, the potential functions for both teams are identical rather than summing to zero, resulting in a potential function that encompasses the individual potential functions. The team-FP algorithm converges to TNE in this case as well (see. Figure 6c). However, the equilibrium is not unique in this case. Finally, we create an extension of the team-FP for Markov games (see. Appendix C) in an RL framework and try simulations on this setting. In this case, there

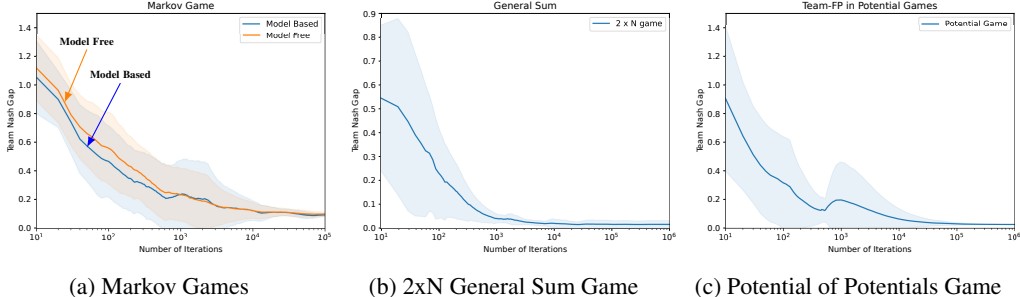

|  (a) Markov Games | (b) 2xN General Sum Game | (c) Potential of Potentials Game |

Figure 6: All the above figures describes the variation of TNG over iterations for Algorithms that are related to but outside the scope of ZSPTG. (a) The model-free and model-based Markov games of Algorithm 2, and 3, for a game of 2-team each with 2 agents, with 2 states and 10 horizon length. (b) The behavior of Team-FP dynamics in a 2xN general sum game, where a team competes against a single agent with random rewards. (c) The behavior of Team-FP dynamics in a potential game over the underlying potential functions.

are two teams each having two agents, competing against each other in a finite horizon Markov game with a horizon length of ten. The number of states is two, and the state transition matrices are generated randomly. We observe that team-Nash Gap for MG's defined in (65), converges to near-zero.

## 6    Conclusion

In this paper, we introduced Team-FP, a novel fictitious play variant designed for multi-team games. We showed that Team-FP provably achieves near-TNE in ZSPTGs, with a quantifiable error bound. Our convergence analysis addressed the non-stationarity challenge arising from evolving opponent team strategies, by leveraging the optimal coupling lemma and stochastic differential inclusion approximation methods. We also extended Team-FP to multi-team Markov games, encompassing both model-based and model-free scenarios, with applications in multi-team reinforcement learning. Furthermore, we conducted detailed numerical analysis of the Team-FP dynamics, focusing on the trade-off in learning to team up and competition in comparison to the classical FP and no-regret learning dynamics. We further examined the convergence of Team-FP dynamics to TNE in multi-team games beyond ZSPTGs. These results strengthen the theoretical foundations for applying TNE to predict decentralized team behavior and provide a framework for team learning in multi-team settings.

**Limitations and Broader Impacts**    Our work quantified the almost sure convergence of the Team-FP dynamics asymptotically yet did not provide guarantees for the convergence rate. We conducted detailed numerical examples and presented the evolution of the gap in TNE to exemplify this rate qualitatively. The challenge for the non-asymptotic analysis is inherited from the *discrete-time* FP dynamics for which there are only rough rate analysis [Karlin, 1959] and negative examples for some edge cases [Brandt et al., 2010, Daskalakis and Pan, 2014].

Our work introduces no new ethical concerns in multi-team systems and shares the assumption of stationary opponents with learning dynamics like Q-learning and fictitious play. This assumption does not disadvantage our approach. We argue that treating uncoordinated attackers as a single decision-maker is necessary, as they can learn to bypass security measures. Our paper provides a theoretical basis for this, ensuring more reliable AI-based solutions.

**Future Research Directions**    This work paves the way to further explore the behavior of decentralized teams in multi-team interactions when the team members follow different types of dynamics for teaming up within teams (other than log-linear learning) and adapting to other teams' play (other than FP). Numerical examples we conducted for multi-team games beyond ZSPTGs are also promising to show the provable convergence of Team-FP dynamics in multi-team (Markov) games reducing to games with the *fictitious play property* (e.g., see [Monderer and Shapley, 1996b]) if the teams coordinate in acting as a single player.

## Acknowledgements

This work was supported by The Scientific and Technological Research Council of Türkiye (TUBITAK) BIDEB 2232-B International Fellowship for Early Stage Researchers under Grant Number 121C124.

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

# A   Proof of Theorem 4.2

We can separate the proof into two main steps: $(i)$ showing that team members can learn to team up approximately by constructing a reference scenario where beliefs got frozen, and $(ii)$ addressing the bounded approximation error by leveraging the stochastic differential inclusion approximations.

## A.1   Step $(i)$ - Reference Scenario and Error Analysis

We divide the horizon into $T$-length epochs. Then, we can write the belief update (8) accumulated from $k = nT$ to $(n+1)T$ as

$$\pi^m_{(n+1)T} = \left( \sum_{k=nT}^{(n+1)T-1} (1 - \alpha_k) \right) \cdot \pi^m_{nT} + \sum_{k=nT}^{(n+1)T-1} \alpha_k \left( \prod_{\ell=k+1}^{k_0+T-1} (1 - \alpha_\ell) \right) \cdot \underline{a}^m_k \tag{16}$$

for any $n = 0, 1, \dots$ denoting the epoch index. Let $\pi^m_{(n)} := \pi^m_{nT}$ denote the belief about team $m$ in epoch $n$. Furthermore, for notational simplicity, we define

$$\beta_k := \alpha_k \prod_{\ell=k+1}^{(n+1)T-1} (1 - \alpha_\ell) \quad \text{and} \quad \beta_{(n)} := \sum_{k=nT}^{(n+1)T-1} \beta_k. \tag{17}$$

Then, we can simplify (16) as

$$\pi^m_{(n+1)} = (1 - \beta_{(n)}) \cdot \pi^m_{(n)} + \beta_{(n)} \left( \sum_{k=nT}^{(n+1)T-1} \frac{\beta_k}{\beta_{(n)}} \cdot \underline{a}^m_k \right). \tag{18}$$

Due to 4.1, the step size $\beta_{(n)}$ decays to zero monotonically at a certain rate such that [Donmez et al., 2024, Lemma 5.4]

$$\sum_{n=0}^{\infty} \beta_{(n)} = \infty \quad \text{and} \quad \sum_{n=0}^{\infty} \beta^2_{(n)} < \infty. \tag{19}$$

Let $\mathcal{F}_{(n)}$ be a filtration generated by the $\sigma$-algebra $\sigma(A_0, \dots, A_{nT-1})$, where $A_t$ denotes the joint actions taken by each team at stage $t$, i.e., $A_t = (\underline{a}^1_t, \dots, \underline{a}^{|\mathcal{T}|}_t)$. It is instructive to note that $\pi^m_{(n)}$ is $\mathcal{F}_{(n)}$-measurable. We define the joint action distributions of team $m$ at time $k$ in epoch $n$ by

$$\mu^m_{(n),k} := \mathrm{E}[\underline{a}^m_k \mid \mathcal{F}_{(n)}]. \tag{20}$$

Then, we can write (18) as in the form of stochastic approximation

$$\pi^m_{(n+1)} = (1 - \beta_{(n)}) \pi^m_{(n)} + \beta_{(n)} \left( \sum_{k=nT}^{(n+1)T-1} \frac{\beta_k}{\beta_{(n)}} \cdot \mu^m_{(n),k} + \omega^m_{(n+1)} \right), \tag{21}$$

where $\omega^m_{(n+1)}$ is a Martingale difference sequence defined by

$$\omega^m_{(n+1)} := \sum_{k=nT}^{(n+1)T-1} \frac{\beta_k}{\beta_{(n)}} \left( \underline{a}^m_k - \mu^m_{(n),k} \right). \tag{22}$$

Now, let's consider a *reference scenario* for the analysis in which the beliefs (denoted by $\widehat{\pi}^m_t$) are only updated at the ends of $T$-length epochs. In other words, we have $\widehat{\pi}^m_t = \pi^m_{(n)}$ for all $nT \leq t \leq (n+1)T - 1$ and $m \in \mathcal{T}$. Due to the fixed beliefs about the opponent plays, Team-FP dynamics reduce to the log-linear learning in the reference scenario. Let $\widehat{a}^m_{(n),k}$ denote the joint action of team $m$ in the reference scenario. By the nature of the log-linear learning, $\{\widehat{a}^m_{(n),k}\}^{\infty}_{k=nT}$ form a homogeneous Markov chain (MC) even though the actual action profiles $\{a^m_k\}^{\infty}_{k=nT}$ do not necessarily do so. Denote the stationary distribution of the MC in the reference scenario by $\breve{\mu}^m_{(n),\star}$ and $\widehat{\mu}^m_{(n),\star}$ for Team-FP and Independent Team-FP, respectively. The former is given by

$\breve{\mu}^m_{(n),\star} = \mathrm{br}_\tau(\phi^m(\cdot, \pi^{-m}_{(n)}))$ due to the log-linear learning update [Blume, 1993, Marden and Shamma, 2012]. Then, we can write the belief update (21) for team $m$ as

$$\pi^m_{(n+1)} = (1 - \beta_{(n)})\pi^m_{(n)} + \beta_{(n)}\left(\mathrm{br}_\tau(\phi^m(\cdot, \pi^{-m}_{(n)})) + \omega^m_{(n+1)} + e^m_{(n)}\right), \tag{23}$$

where we decompose the error $e^m_{(n)} := \widehat{e}^m_{(n)} + \breve{e}^m_{(n)}$ as

$$\widehat{e}^m_{(n)} := \sum_{k=nT}^{(n+1)T-1} \frac{\beta_k}{\beta_{(n)}}(\mu^m_{(n),k} - \widehat{\mu}^m_{(n),\star}) \tag{24}$$

$$\breve{e}^m_{(n)} := \widehat{\mu}^m_{(n),\star} - \breve{\mu}^m_{(n),\star}. \tag{25}$$

The update (23) corresponds to the SFP dynamics of teams acting as a single decision-maker with the additive error $e^m_{(n)}$.

The following lemma, based on designing the optimal coupling between the actual and reference scenarios as in [Donmez et al., 2024, Lemma 5.5], plays an important role in bounding $\|\widehat{e}^m_{(n)}\|$ from above.

**Lemma A.1.** *For some constants $c, d, \rho \geq 0$, we have*

$$\|\mu^m_{(n),k} - \widehat{\mu}^m_{(n),\star}\|_1 \leq c\,\rho^{k-nT} + d\,T\alpha_{nT} \quad \forall m \in \mathcal{T}. \tag{26}$$

By (24), Lemma A.1 yields that we can bound the error $\widehat{e}^m_{(n)}$ from above by

$$\|\widehat{e}^m_{(n)}\|_1 \leq c \sum_{k=nT}^{(n+1)T-1} \frac{\beta_k}{\beta_{(n)}}\rho^{k-nT} + dT\alpha_{nT}. \tag{27}$$

Due to the no-recency-bias condition in 4.1, we have $\beta_{k+1}/\beta_k \leq 1$ by the definition (17). Then, we have monotonically decaying $\beta_k$, which yields

$$\frac{\beta_k}{\beta_{(n)}} \leq \frac{\beta_{nT}}{T\beta_{(n+1)T-1}} \leq \frac{\alpha_{nT}}{T\alpha_{(n+1)T}}. \tag{28}$$

By the assumption 4.1, we have

$$\lim_{n\to\infty} \frac{\alpha_{nT}}{T\alpha_{(n+1)T}} = \frac{1}{T}\lim_{n\to\infty} \prod_{k=nT}^{(n+1)T-1} \frac{\alpha_k}{\alpha_{k+1}} = \frac{1}{T}. \tag{29}$$

By (27), (28), (29), and the decaying property of $\alpha_k$, we obtain

$$\limsup_{n\to\infty} \|\widehat{e}^m_{(n)}\|_1 \leq \frac{c}{T}\frac{1}{1-\rho} \quad \forall m \in \mathcal{T}. \tag{30}$$

On the other hand, $\breve{e}^m_{(n)}$ is non-zero only for Independent Team-FP and corresponds to the difference between the stationary distributions for the classical and independent log-linear learning. We can bound $\breve{e}^m_{(n)} \leq \Lambda(\delta, \epsilon)$ for some $\epsilon > 0$ based on (13). Hence, given any $\varepsilon > 0$, there exists $N_\varepsilon$ such that

$$\|e^m_{(n)}\|_1 < C(\varepsilon, T) \quad \forall n \geq N_\varepsilon, \tag{31}$$

where

$$C(\varepsilon, T) := \begin{cases} \varepsilon + \frac{c}{T}\frac{1}{1-\rho} & \text{for Team-FP} \\ \varepsilon + \frac{c}{T}\frac{1}{1-\rho} + \Lambda(\delta, \epsilon) & \text{for Independent Team-FP} \end{cases} \tag{32}$$

Note that $C(\varepsilon, T)$ can become arbitrarily close to $\Lambda(\delta, \epsilon)$ for sufficiently large $T$ and small $\varepsilon$, which are chosen arbitrarily just for the analysis.

## A.2 Step $(ii)$ - Convergence Analysis with Relaxed Errors

We focus on the convergence analysis of (23) based on the bound (31). To this end, we define the set-valued mapping

$$F(\pi) := \Big\{ \big(\mathrm{br}_\tau(\phi^m(\cdot, \pi^{-m})) - \pi^m + e^m\big)_{m \in \mathcal{T}} :$$

$$\|e^m\|_1 \leq C(\varepsilon, T) \text{ and } \mathrm{br}_\tau(\phi^m(\cdot, \pi^{-m})) + e^m \in \Delta(\underline{A}^m) \ \forall m \Big\} \qquad (33)$$

for all $\pi \in \Pi := \prod_{m \in \mathcal{T}} \Delta(\underline{A}^m)$. Then, the update (23) and (31) yield that for sufficiently large $n$, the empirical averages $\{\pi_{(n)}\}_{n \geq 0}$ satisfy

$$\pi_{(n+1)} - \pi_{(n)} - \beta_{(n)} \cdot \omega_{(n+1)} \in \beta_{(n)} \cdot F(\pi_{(n)}), \qquad (34)$$

where the Martingale difference sequence $\{\omega_{(n+1)}\}$ is as described in (22). We have set inclusion in (34) different from the equality in (23) since we relax the update rule by considering any error satisfying the bound (31), rather than the actual error.

The following proposition shows that $F(\cdot)$ is a peculiar set-valued mapping. Particularly, given the sets $X, Y$ from some underlying Euclidean space, we say that a set-valued function $\overline{F}(\cdot)$ mapping each point $x \in X$ to a set $\overline{F}(x) \subset Y$ is a *Marchaud map*, e.g., see [Perkins and Leslie, 2013, Definition 2.1] and [Benaïm et al., 2005, Hypothesis 1.1], if it satisfies the following conditions:

(i) $\overline{F}(\cdot)$ is upper semi-continuous, or equivalently, $\mathrm{Graph}(\overline{F}) = \{(x, y) : y \in \overline{F}(x)\}$ is a closed subset of $X \times Y$.

(ii) For all $x \in X$, the set $\overline{F}(x)$ is a non-empty, compact, and convex subset of $Y$.

(iii) There exists a $c > 0$ such that $\sup_{y \in \overline{F}(x)} \|y\| \leq c(1 + \|x\|)$ for all $x \in X$.

**Proposition A.2.** *The set-valued function $F(\cdot)$ is a Marchaud map.*

Next, we approximate the relaxation (34) by the differential inclusion

$$\dot{\pi} \in F(\pi). \qquad (35)$$

Particularly, due to Proposition A.2 and (19), a linear interpolation of the iterative process $\{\pi_{(n)}\}_{n \geq 0}$ is a perturbed solution to the differential inclusion (35) [Benaïm et al., 2005, Proposition 1.4] and its limit set is internally chain transitive [Benaïm et al., 2005, Theorem 3.6]. Furthermore, we can characterize internally chain transitive sets (and therefore, the limit set of linear interpolations) through a Lyapunov function. Particularly, let $\Phi_t(\pi)$ be the set of solutions to (35) with initial point $\pi$. We say that a continuous function $\overline{V} : \Pi \to \mathbb{R}$ is a *Lyapunov function* of (35) for a subset $\Lambda \subset \Pi$ provided that

- $\overline{V}(\tilde{\pi}) < \overline{V}(\pi)$ for all $\pi \in \Pi \setminus \Lambda$, $\tilde{\pi} \in \Phi_t(\pi)$, $t > 0$,
- $\overline{V}(\tilde{\pi}) \leq \overline{V}(\pi)$ for all $\pi \in \Lambda$, $\tilde{\pi} \in \Phi_t(\pi)$, $t \geq 0$.

Given such a Lyapunov function for $\Lambda$, every chain internally chain transitive set $L$ is contained in $\Lambda$ if the set $\{\overline{V}(\pi) : \pi \in \Lambda\}$ has empty interior [Benaïm et al., 2005, Proposition 3.27]. Therfore, we propose the candidate Lyapunov function

$$V(\pi) = \max\{0, L(\pi) - \Xi(\lambda, \varepsilon, T)\}, \qquad (36)$$

where the auxiliary terms are defined by

$$L(\pi) := \sum_{m \in \mathcal{T}} \max_{\mu \in \Delta(\underline{A}^m)} \big\{\phi^m(\mu, \pi^{-m}) + \tau \mathcal{H}(\mu)\big\} \qquad (37a)$$

$$\Xi(\lambda, \varepsilon, T) := \tau \log|A| + \lambda |\mathcal{T}|^2 \overline{\phi} \cdot C(\varepsilon, T) > 0 \qquad (37b)$$

for some arbitrary $\lambda > 1$ to characterize the long-run behavior of (34) based on (35).

**Proposition A.3.** *The continuous function $V(\cdot)$ is a Lyapunov function of (35) for the set $\Lambda = \{\pi : V(\pi) = 0\}$.*

Proposition A.3 yields that there exists some sequence $\{\zeta_k \in \mathbb{R}\}_{k \geq 0}$ such that $|\zeta_k| \to 0$ as $k \to \infty$ almost surely and

$$L(\pi_k) < \Xi(\lambda, \varepsilon, T) + \zeta_k \quad \forall k. \tag{38}$$

By (3), we can write $L(\pi_k)$ as

$$L(\pi_k) = \sum_{m \in \mathcal{T}} \left( \max_{\mu \in \Delta(\underline{A}^m)} \left\{ \phi^m(\mu, \pi_k^{-m}) + \tau \mathcal{H}(\mu) \right\} - \phi^m(\pi_k) \right),$$

$$\geq \sum_{m \in \mathcal{T}} \left( \max_{\mu \in \Delta(\underline{A}^m)} \left\{ \phi^m(\mu, \pi_k^{-m}) \right\} - \phi^m(\pi_k) \right). \tag{39}$$

Since $\max_\mu \left\{ \phi^m(\mu, \pi^{-m}) \right\} - \phi^m(\pi) \geq 0$ for all $\pi$, the inequalities (38) and (39) yield that

$$0 \leq \sum_{m \in \mathcal{T}} \left( \max_{\mu \in \Delta(\underline{A}^m)} \left\{ \phi^m(\mu, \pi_k^{-m}) \right\} - \phi^m(\pi_k) \right) \leq \Xi(\lambda, \varepsilon, T) + \zeta_k \; \forall m \text{ and } n. \tag{40}$$

Recall that we can select $\lambda > 1$, $\varepsilon > 0$ and $T \in \mathbb{N}$ arbitrarily. The definitions (32) and (37b) yield that $\Xi(\lambda, \varepsilon, T)$ can be arbitrarily close to $\tau \log |A|$ for Team-FP and $\tau \log |A| + |\mathcal{T}|^2 \overline{\phi} \cdot \Lambda(\delta, \epsilon)$ for Independent Team-FP. Furthermore, $\zeta_k$ decays to zero. Therefore, we can obtain (14) based on (40) and Definition 2.4. This completes the proof.

# B    Proofs of Technical Results

The followings are the proofs of Lemma A.1, and Propositions A.2 and A.3 used in Appendix A.

## B.1    Proof of Lemma A.1

We define two discrete-time processes from the beginning of epoch $n$. One of them is the joint actions of teams in the reference scenario, defined as $\{\widehat{\omega}_k\}_{k \geq nT} := \{(\widehat{a}_k^m)_{m \in \mathcal{T}} | \mathcal{F}_{(n)}\}_{k \geq nT}$, and the other one is the joint actions of teams in the actual scenario, with $\{\omega_k\}_{k \geq nT} := \{(a_k^m)_{m \in \mathcal{T}} | \mathcal{F}_{(n)}\}_{k \geq nT}$.

The transition probabilities between states depend only on the beliefs on the other teams. Therefore, for the reference scenario, during an epoch, this process is a homogeneous MC. Let, $P_{(n),k}$ be the transition probabilities between the states, i.e., joint action profiles of all teams, of the original discrete-time process at step k, and let $\widehat{P}_{(n)}$ be the transition probabilities between the states for the reference scenario. Let $\widehat{\mu}_{(n),k}$ be the distribution of $\widehat{\omega}_k$, and let $\mu_{(n),k}$ be the distribution of $\omega_k$. In this case, the expected joint actions of a team at step k within an epoch in real and fictional scenarios can be expressed in terms of the distributions as:

$$\mu_{(n),k}^m(a) = \sum_{\omega : \{a^m = a\}} \mu_{(n),k}(\omega), \qquad \widehat{\mu}_{(n),k}^m(a) = \sum_{\widehat{\omega} : \{a^m = a\}} \widehat{\mu}_{(n),k}(\widehat{\omega}). \tag{41}$$

**Step 1.** In classical log-linear learning, at each step, only one agent can change their action and due to the soft-max nature of this action change, any action has positive probability which is bounded from below. This bound only depends on the minimum and maximum values of any agents' reward, which are defined by the game and bounded by definition, and the temperature parameter. If we divide that bound by the number of agents in the team ($|\mathcal{I}^m|$), we can obtain a lower bound on the probability of changing to any joint action which can be reached within a single step. Let's call that bound $\xi$. Then, for any state $\omega$ or $\widehat{\omega}$, there is a $\xi > 0$ such that

$$P_{(n),k}(\omega|\omega_k) > \xi^{|\mathcal{T}|} \iff P_{(n),k}(\omega|\omega_k) > 0, \tag{42}$$

$$\widehat{P}_{(n),k}(\widehat{\omega}|\widehat{\omega}_k) > \xi^{|\mathcal{T}|} \iff \widehat{P}_{(n),k}(\widehat{\omega}|\widehat{\omega}_k) > 0. \tag{43}$$

Then, we can find a path with positive probability for both of these scenarios, where their state does not match until they reach $\kappa = \max_m |\mathcal{I}^m|$. In other words, $\omega_{nT+\kappa} = \widehat{\omega}_{nT+\kappa}$ and $\omega_k \neq \widehat{\omega}_k$ for all $k = nT, \ldots, nT + \kappa - 1$ with

$$\Pr\left( \bigwedge_{k=nT+1}^{nT+\kappa} \widehat{\omega}_k \mid \widehat{\omega}_{nT} \right) \geq \xi^\kappa \quad \text{and} \quad \Pr\left( \bigwedge_{k=nT+1}^{nT+\kappa} \omega_k \mid \omega_{nT} \right) \geq \xi^\kappa. \tag{44}$$

Hence, (44) satisfies the first condition for [Donmez et al., 2024, Lemma 2].

**Step 2.** For this part, we introduce a notation $a^{jm} \in A^{jm}$, where $jm$ represents the $j^{th}$ agent from team $m$. For example, if all teams have identical agent numbers and agent indexes are ordered for teams, $jm = (m-1)|T_m| + j$. Also, let $\pi_{(n),k} := (\pi^m_{(n),k})_{m \in \mathcal{T}}$, and $\widehat{\pi}_{(n),k} := (\widehat{\pi}^m_{(n),k})_{m \in \mathcal{T}}$. Now, consider the total variation distance between two transition probabilities. Transition probabilities between state $\omega = \{(a^{im}, a^{-im})\}_{m \in \mathcal{T}}$ and $\widetilde{\omega} = \{(\widetilde{a}^{im}, a^{-im})\}_{m \in \mathcal{T}}$, where $im$ can be any random agent from team $m$ with a slight abuse of notation, can be written as a function of the belief as

$$P_{\omega \to \widetilde{\omega}}(\pi_{(n),k}) = \prod_{m \in \mathcal{T}} P^m_{\omega \to \widetilde{\omega}}(\pi_{(n),k}), \tag{45}$$

where $P^m_{\omega \to \widetilde{\omega}}$ is defined to be

$$P^m_{\omega \to \widetilde{\omega}}(\pi_{(n),k}) = \frac{1}{|\mathcal{I}^m|} \frac{\exp\left(\left(\phi^m\left(\widetilde{a}^{im}, a^{-im}, \pi^{-m}_{(n),k}\right)\right)/\tau\right)}{\sum\limits_{\widetilde{a}' \in A^{im}} \exp\left(\phi^m\left(\widetilde{a}', a^{-im}, \pi^{-m}_{(n),k}\right)/\tau\right)}, \tag{46}$$

and $\pi^{-m}_{(n),k} := (\pi^\ell_{(n),k})_{\ell \neq m}$. Remember that due to the separable structure of the network in ZSPTG we have

$$\phi^m(a^{im}, (a^{-im}), \pi^{-m}_{(n),k}) = \sum_{\ell \neq m} \mathrm{E}_{\underline{a}^\ell \sim \pi^\ell_{(n),k}} \phi^{m\ell}(a^{im}, a^{-im}, \underline{a}^\ell) \tag{47a}$$

$$= \sum_{\ell \neq m} (\underline{a}^m)^T \Phi^{m\ell} \pi^\ell_{(n),k}, \tag{47b}$$

where $\Phi^{m\ell}$ is the matrix form of the potential function whose rows are the joint actions of team $m$ and columns are the joint actions of team $\ell$.

Now, for all $\omega_{k+1}$, which is reachable in one step from the state $\omega_k$, i.e., at most a single agent changes action in each team, the relation between state transition probabilities and $P^m_{\omega \to \widetilde{\omega}}(\cdot)$ can be expressed as

$$P_{(n),k}(\omega_{k+1}|\omega_k, \dots, \omega_{nT}) = \prod_{m \in \mathcal{T}} P^m_{\omega_k \to \omega_{k+1}}(\pi_{(n),k}), \tag{48a}$$

$$\widehat{P}_{(n)}(\omega_{k+1}|\omega_k) = \prod_{m \in \mathcal{T}} P^m_{\omega_k \to \omega_{k+1}}(\widehat{\pi}_{(n),k}). \tag{48b}$$

Note that $\pi_{(n),k}$ is a function of history of actions and the initial $\pi_{(n)}$. Therefore, it can be computed given the past states $(\omega_k, \dots, \omega_{nT})$.

We know that $P^m_{\omega \to \widetilde{\omega}}(\pi_{(n),k})$ is bounded as it is a probability. Now, using the Lipschitz property of the soft-max function, and since $\phi^m(\cdot, \pi^{-m}_{(n),k})$ is a linear function of $\pi_{(n),k}$, we can say that $P^m_{\omega_k \to (\cdot)}(\pi)$ is a Lipschitz continuous function with respect to the input $\pi$. Also, using the fact that the product of bounded Lipschitz continuous functions is also Lipschitz continuous, and (48), we can say that there exists an $\mathcal{L} < \infty$ such that the total variation distance between two transition probabilities is bounded as

$$\left\| P_{(n),k}(\cdot|\omega_k, \dots, \omega_{nT}) - \widehat{P}_{(n)}(\cdot|\omega_k) \right\|_{\mathrm{TV}} \leq \mathcal{L} \sum_{m \in \mathcal{T}} \left\| (\pi^\ell_{(n),k} - \widehat{\pi}^\ell_{(n),k}) \right\|_1. \tag{49}$$

The distance between the belief of the original scenario and the reference scenario can also be bounded thanks to the small step size. If we bound $\|a^\ell_k - \pi^\ell_k\|_1 < \|a^\ell_k\|_1 + \|\pi^\ell_k\|_1 = 2$. Then using triangle inequality

$$\left\| (\pi^\ell_{(n),k} - \widehat{\pi}^\ell_{(n),k}) \right\|_1 = \left\| (\pi^\ell_{(n),k} - \pi^\ell_{(n),nT}) \right\|_1 \leq \sum_{t=nT}^{k+NT-1} 2\alpha_t \tag{50a}$$

$$\leq \sum_{t=nT}^{(n+1)T-1} 2\alpha_t \tag{50b}$$

$$\leq 2T\alpha_{nT}. \tag{50c}$$

Let's consider late epochs where $\alpha_{nT} < \dfrac{1}{2T\mathcal{L}|\mathcal{T}|}$, and set $2T\mathcal{L}|\mathcal{T}|\alpha_{nT} = 1 - \lambda_{nT}$ with $0 < \lambda_{nT} \le 1$. Then,

$$\left\| P_{\omega \to \widetilde{\omega}}(\pi_{(n),k}^{-m}) - P_{\omega \to \widetilde{\omega}}(\widehat{\pi}_{(n),k}^{-m}) \right\|_{\mathrm{TV}} \le 1 - \lambda_{nT}. \tag{51}$$

Hence, the second condition of [Donmez et al., 2024, Lemma 2] is met, and we can invoke the corresponding Lemma such that given the distributions of the original and reference scenarios, following inequality holds for all $k \ge nT$,

$$\left\| \mu_{(n),k} - \widehat{\mu}_{(n),k} \right\|_1 \le 2(1-\varepsilon)^{\frac{k-nT}{\kappa}-1} + 2\left(1 - \lambda_{nT}^{\kappa}\right)\frac{1+\varepsilon}{\varepsilon}, \tag{52}$$

where $\varepsilon = \xi^{2\kappa}$. If we define constants, $c := 2(1-\varepsilon)^{\frac{1}{k}-1}$, $\rho := (1-\varepsilon)^{\frac{1}{\kappa}}$, $d := 4\dfrac{1+\varepsilon}{\varepsilon}$, and assume that the reference scenario initial distribution is the stationary distribution of the MC, $\widehat{\mu}_{(n),\star}$, we can rewrite the inequality (52) as follows

$$\left\| \mu_{(n),k} - \widehat{\mu}_{(n),\star} \right\|_1 \le c \cdot \rho^{k-nT} + d \cdot T\alpha_{nT}, \tag{53}$$

for all $k \ge nT$. Then, by (41), and triangle inequality, we can conclude

$$\left\| \mu_{(n),k}^m - \widehat{\mu}_{(n),\star}^m \right\|_1 \le \left\| \mu_{(n),k} - \widehat{\mu}_{(n),\star} \right\|_1 \le c \cdot \rho^{k-nT} + d \cdot T\alpha_{nT}, \tag{54}$$

for all $k \ge nT$.

## B.2 Proof of Proposition A.2

The set $\Pi$ is compact set by definition as it is a Cartesian product of probability simplexes. Let's consider a convergent sequence $(\pi_n, \mu_n - \pi_n + e_n)_{n=1,2,\dots}$ in the set $\{(\pi_n, y) : y \in F(\pi)\}$, and let $(\pi^\star, \mu^\star - \pi^\star + e^\star)$ be the point that the sequence converges to. Given $\pi_n$, any $\mu_n$ is a fixed and unique value, and it is an element of the compact set that is generated by mapping probability simplex with the continuous soft-max function. Then, for any $\pi^\star \in \Pi$, $\mu^\star - \pi^\star$ is a fixed value within another compact set. Furthermore, the error term must remain within the compact set $e^\star \in e$. As a result, $(\pi^\star, \mu^\star - \pi^\star + e^\star)$ is also within the set $\{(\pi_n, y) : y \in F(\pi)\}$, and $F : \Pi \to A^{\sum_{m \in \mathcal{T}} \|\underline{A}^m\|}$ is a closed-set valued map. Therefore, the condition (i) is satisfied. Given a $\pi \in \Pi$, $\mu_\star \in \Pi$ is a fixed value corresponding to the smoothed best responses to $\pi_{\{m \in \mathcal{T}\}}^{-m}$. Hence, $\mu_\star - \pi$ is a fixed value for a given $\pi$. Note that each $e^m \in e$ is a non-empty, bounded, closed and convex subset of $\mathbb{R}^{\sum_{m \in \mathcal{T}} \|\underline{A}^m\|}$. Therefore, for any given $\pi \in \Pi$, $F(\pi) = \mu_\star - \pi + e$ is a non-empty, compact, convex subset of $\mathbb{R}^{\sum_{m \in \mathcal{T}} \|\underline{A}^m\|}$. As a result, (ii) is also satisfied. The function $F$ is bounded such that

$$\sup_{y \in F(x)} \|y\|_1 \le \sup_{\pi \in \Delta} \|\pi\|_1 + \sup_{\mu \in \Delta} \|\mu\|_1 + \sup \|e\| \le 2M + M\left(\frac{1}{T}\frac{1}{1-\rho} + K^m(\delta)\right). \tag{55}$$

Hence, it satisfies the condition (iii). Since all three conditions are satisfied, $F$ is a Marchaud Map.

## B.3 Proof of Proposition A.3

The smoothness of the entropy regularization in (1) yields that we can invoke the envelope theorem to compute the time derivative of $L(\pi)$ as

$$\frac{d}{dt}L(\pi) = \sum_{m \in \mathcal{T}} \phi^m(\mu_\star^m, \dot{\pi}^{-m}) \tag{56}$$

$$\overset{(a)}{=} \sum_{m \in \mathcal{T}} \sum_{\ell \ne m} \phi^{m\ell}(\mu_\star^m, \dot{\pi}^\ell), \tag{57}$$

where $\mu_\star^m := \mathrm{br}_\tau(\phi^m(\cdot, \pi^{-m}))$ and $(a)$ follows from (4). By (35) and (33), we have $\dot{\pi}^\ell = \mu_\star^\ell - \pi^\ell + e^\ell$. Therefore, we can write (57) as

$$\frac{d}{dt}L(\pi) = \sum_{m \in \mathcal{T}} \sum_{\ell \ne m} \left(\phi^{m\ell}(\mu_\star^m, \mu_\star^\ell) - \phi^{m\ell}(\mu_\star^m, \pi^\ell) + \sum_{\underline{a}^\ell \in \underline{A}^\ell} e^\ell(\underline{a}^\ell)\phi^{m\ell}(\mu_\star^m, \underline{a}^\ell)\right) \tag{58}$$

For the first two terms on the right-hand side, we have

$$\sum_{m \in \mathcal{T}} \sum_{\ell \neq m} \left( \phi^{m\ell}(\mu_\star^m, \mu_\star^\ell) - \phi^{m\ell}(\mu_\star^m, \pi^\ell) \right) \overset{(a)}{=} - \sum_{m \in \mathcal{T}} \phi^m(\mu_\star^m, \pi^{-m}), \tag{59}$$

$$\overset{(b)}{\leq} -L(\pi) + \tau \log |A|, \tag{60}$$

where $(a)$ is due to (3) and (4), and $(b)$ follows from the definition (37a) as $\mathcal{H}(\mu_\star^m) \leq \log |\underline{A}^m|$. On the other hand, we have

$$\sum_{m \in \mathcal{T}} \sum_{\ell \neq m} \sum_{\underline{a}^\ell \in \underline{A}^\ell} e^\ell(\underline{a}^\ell) \phi^{m\ell}(\mu_\star^m, \underline{a}^\ell) \leq \sum_{m \in \mathcal{T}} \sum_{\ell \neq m} \|e^\ell\|_1 \cdot \overline{\phi} \leq |\mathcal{T}|^2 \overline{\phi} \cdot C(\varepsilon, T), \tag{61}$$

due to the bound on the errors. By the bounds (60) and (61), we can bound the time derivative of $L(\pi)$ as

$$\frac{d}{dt} L(\pi) < -L(\pi) + \Xi(\lambda, \varepsilon, T) \Leftrightarrow \frac{d}{dt} \left( L(\pi) - \Xi(\lambda, \varepsilon, T) \right) < -L(\pi) + \Xi(\lambda, \varepsilon, T), \tag{62}$$

where the constant $\Xi(\lambda, \varepsilon, T) > 0$ is as described in (37b). Since we have $V(\pi) = \min(0, L(\pi) - \Xi(\lambda, \varepsilon, T))$, the strict inequality in (62), yields that $V(\cdot)$ is a Lyapunov function.

# C  Extension to Multi-team Markov Games

Markov games (MGs), introduced by Shapley [1953], generalizes Markov decision processes (MDPs) to non-cooperative multi-agent environments. We can characterize a multi-team MG by the tuple $\langle H, \mathcal{T}, S, (A^i, r^i)_{i \in \mathcal{I}}, p, p_o \rangle$, where $H$ is the horizon length, $\mathcal{T}$ and $\mathcal{I}$ again denote the index sets for the teams and agents, $S$ denotes the *finite* set of states, and $A^i$ and $r^i : S \times A \to \mathbb{R}$ denote, resp., the agent $i$'s finite action set and *immediate reward* function, depending on current state and joint actions.[1] The state of the underlying game can change according to the transition kernel $p(\cdot \mid s, a)$, depending on the current state and joint actions, and the initial state is determined by the probability distribution $p_o \in \Delta(S)$.

Let each team $m$ randomize their actions contingent on the current state $s \in S$ and stage $h \in [H] := \{1, \ldots, H\}$ via a stationary strategy $\underline{\pi}^m : S \times [H] \to \Delta(\underline{A}^m)$. Note that team members do not necessarily randomize their actions independently. Given the strategy profile of teams $\underline{\pi} = (\underline{\pi}^m)_{m \in \mathcal{T}}$, agent $i$'s utility function is defined by

$$U^i(\underline{\pi}) := \mathrm{E} \left[ \sum_{h=1}^{H} r^i(s_h, a_h) \right], \tag{63}$$

where the pair $(s_h, a_h)$ denotes the state and action profile at stage $h$, the expectation is taken with respect to the randomness on these pairs $(s_h, a_h)$ induced by the strategy profile $\underline{\pi}$ and the underlying transition kernel.

For each state $s$, let the reward functions $\{r^i(s, \cdot)\}_{i \in \mathcal{I}}$ induce a ZSPTG where team $m \in \mathcal{T}$ has the potential function $\phi^m(s, \cdot) : A \to \mathbb{R}$ satisfying (2), (3), and (4). Correspondingly, team $m$'s utility function is given by

$$\underline{U}^m(\underline{\pi}) := \mathrm{E} \left[ \sum_{h=1}^{H} \phi^m(s_h, a_h) \right]. \tag{64}$$

Let $\underline{\Pi}^m := \{\underline{\pi}^m \mid \underline{\pi}^m : S \times [H] \to \Delta(\underline{A}^m)\}$ denote the set of stationary strategy profiles for team $m$. Then, the following is the counterpart of Definition 2.4 for multi-team MGs.

**Definition C.1** (Team-Nash Gap for MGs). Given the strategy profile of teams $\{\underline{\pi}^m \in \underline{\Pi}^m\}_{m \in \mathcal{T}}$, we define the *team-Nash gap* for team $m$ as

$$\underline{\mathrm{TNG}}^m(\pi) := \max_{\widetilde{\underline{\pi}} \in \underline{\Pi}^m} \left\{ \underline{U}^m(\widetilde{\underline{\pi}}, \underline{\pi}^{-m}) \right\} - \underline{U}^m(\underline{\pi}), \tag{65}$$

and $\underline{\mathrm{TNG}}(\pi) := \sum_{m \in \mathcal{T}} \underline{\mathrm{TNG}}^m(\pi)$, where $\underline{\pi}^{-m} := \{\underline{\pi}^\ell\}_{\ell \neq m}$. Correspondingly, we say that the strategy profile of teams $\{\underline{\pi}^m\}_{m \in \mathcal{T}}$ is $\epsilon$-*TNE* if $\underline{\mathrm{TNG}}(\pi) \leq \epsilon$.

---

[1]The results can be generalized to state-variant action sets rather straightforwardly.

Next, we describe the stage-game framework, going back to the introduction of Markov games [Shapley, 1953], and also used in multi-agent reinforcement learning algorithms such as Minimax-Q [Littman, 1994] and Nash-Q [Hu and Wellman, 2003]. Particularly, at each stage, the agents' joint actions determine the immediate rewards they receive and the next state, and therefore, the future rewards to be received. Given the strategy profile of teams $\underline{\pi} = (\underline{\pi}^m)_{m \in \mathcal{T}}$, let $v^i : S \times [H] \to \mathbb{R}$ denote agent $i$'s *value function* such that $v^i(s)$ is the game value for the stage game associated with state $s$. Similarly, let $Q^i : S \times [H] \times A \to \mathbb{R}$ be the *Q-function* such that $Q^i(s, \cdot)$ corresponds to agent $i$'s payoff function for the stage game associated with state $s$. The definition of the utility (63) yields that

$$Q^i(s, h, a) = r^i(s, a) + \mathbb{I}_{\{h < H\}} \sum_{s_+ \in S} p(s_+ \mid s, a) \cdot v^i(s_+, h + 1), \tag{66a}$$

$$v^i(s, h) = Q^i(s, h, \underline{\pi}(s, h, \cdot)). \tag{66b}$$

The Q-function and value function implicitly depend on $\underline{\pi}$.

We can extend Team-FP dynamics to MGs played repeatedly. Based on the stage game framework, agents play some stage game associated with each state $s \in S$ and stage $h \in [H]$ pair repeatedly whenever the underlying MG visits state $s$ at stage $h$. Correspondingly, agents form beliefs about the opponent teams as if the opponent teams play according to some stationary strategies across these repetitions.

Let $\underline{\pi}_k^\ell : S \times [H] \to \Delta(\underline{A}^\ell)$ denote the beliefs formed by agents $i \notin \mathcal{I}^\ell$ about team $\ell$, and $Q_k^i : S \times [H] \times A \to \mathbb{R}$ denote agent $i$'s Q-function estimate at the $k$th repetition. If the underlying MG visits state $s$ at stage $h$ at the $k$th repetition, then agent $i$ follows Team-FP dynamics as if the payoff function is the Q-function estimate $Q_k^i(s, h, \cdot)$. Agents also recall the previous actions of their teams specific to each stage game. We denote agent $i$'s previous action for the pair of state $s$ and stage $h$ until and including the $k$th repetition by $a_k^i(s, h) \in A^i$, with a slight abuse of notation. Then, at stage $k$, agent $i \in \mathcal{I}^m$ either takes the previous action for that stage game (i.e., $a_k^i(s, h) = a_{k-1}^i(s, h)$), or takes the action $a_k^i(s, h) \sim \mathrm{br}_\tau(Q_k^i(s, h, \cdot, a_{k-1}^{-i}(s, h), \underline{\pi}_k^{-m}(s, h)))$ according to the smoothed best response to the previous actions of the team members $a_{k-1}^{-i}(\cdot) := \{a_{k-1}^j(\cdot)\}_{j \in \mathcal{I}^m \setminus \{i\}}$ and the beliefs $\underline{\pi}_k^{-m} := \{\underline{\pi}_k^\ell\}_{\ell \neq m}$ formed about other teams.

Let $s_{h,k}$ and $a_{h,k}$ denote, resp., the state and action profile at stage $h$ at the $k$th repetition. Then, given some reference step size $\{\alpha_c\}_{c \geq 0}$, agents $j \notin \mathcal{I}^m$ update their beliefs about team $m$'s strategy according to

$$\underline{\pi}_{k+1}^m(s, h) = \underline{\pi}_k^m(s, h) + \lambda_k(s, h) \cdot (\underline{a}_k^m(s, h) - \underline{\pi}_k^m(s, h)) \quad \forall k = 0, 1, \ldots, \tag{67a}$$

$$\lambda_k(s, h) = \mathbb{I}_{\{s = s_{h,k}\}} \alpha_{c_k(s,h)}, \tag{67b}$$

where $\underline{a}_k^m(\cdot) := \{a_k^j(\cdot)\}_{j \in \mathcal{I}^m}$ and $c_k(s, h)$ is the number of times state $s$ get visited at stage $h$ until the $k$th repetition. Correspondingly, by (66), each agent $i \in \mathcal{I}^m$ updates their Q-function estimates according to

$$Q_{k+1}^i(s, h, a) = Q_k^i(s, h, a) + \overline{\lambda}_k(s, h, a) \left( \widehat{Q}_k^i(s, h, a) - Q_k^i(s, h, a) \right), \tag{68}$$

where $\overline{\lambda}_k(s, h, a) \in [0, 1]$ is also some step size. If the agent $i$ knows the model of the underlying MG, then we have

$$\widehat{Q}_k^i(s, h, a) = r^i(s, a) + \mathbb{I}_{\{h < H\}} \sum_{s_+ \in S} p(s_+ \mid s, a) \cdot v_k^i(s_+, h + 1), \tag{69a}$$

$$\overline{\lambda}_k(s, h, a) = \mathbb{I}_{\{s = s_{h,k}\}} \alpha_{c_k(s,h)}, \tag{69b}$$

for all $a \in A$ and

$$v_k^i(s, h) = Q_k^i(s, h, \underline{a}_k^m(s, h), \underline{\pi}_k^{-m}(s, h)) \quad \forall (s, h) \in S \times [H]. \tag{70}$$

If the agent does not know the model, then we have

$$\widehat{Q}_k^i(s, h, a) = r_{h,k}^i + \mathbb{I}_{\{h < H\}} v_k^i(s_{h+1,k}, h + 1), \tag{71a}$$

$$\overline{\lambda}_k(s, h, a) = \mathbb{I}_{\{(s,a) = (s_{h,k}, a_{h,k})\}} \alpha_{c_k(s,h,a)}, \tag{71b}$$

---

**Algorithm 2** Model-based (Independent) Team-FP for MGs

---

1: **initialize:** $\{\underline{\pi}_0^\ell(\cdot)\}_{\ell \neq m}$, $\{a_{-1}^j(\cdot)\}_{j \in \mathcal{I}^m \setminus \{i\}}$, and $Q_0^i(\cdot)$ arbitrarily for the typical agent $i \in \mathcal{I}^m$
2: **for** each repetition $k = 0, 1, \ldots$ **do**
3:   **for** each stage $h = 1, \ldots, H$ **do**
4:     **require:** $\{\underline{\pi}_k^\ell(\cdot)\}_{\ell \neq m}$, $\{a_{k-1}^j(\cdot)\}_{j \in \mathcal{I}^m \setminus \{i\}}$, and $Q_k^i(\cdot)$
5:     observe current state $s$
6:     set $\bar{s} = (s, h)$
7:     play $a_k^i(\bar{s}) \sim \mathrm{br}_\tau(Q_k^i(\bar{s}, \cdot, a_{k-1}^{-i}(\bar{s}), \underline{\pi}_k^{-m}(\bar{s})))$ or $a_k^i(\bar{s}) = a_{k-1}^i(\bar{s})$ in a coordinated way (or independently) simultaneously with other agents
8:     observe $a_{h,k}^{-i} = a_k^{-i}(\bar{s})$
9:     receive $r_{h,k}^i = r^i(s, a_{h,k})$
10:   **end for**
11:   **require:** trajectory $\{s_{h,k}, a_{h,k}^{-i}\}_{h=1}^H$
12:   set

$$v_k^i(s, h) = Q_k^i(s, h, \underline{a}_k^m(s, h), \underline{\pi}_k^{-m}(s, h))$$
$$\widehat{Q}_k^i(s, h, \cdot) = r^i(s, \cdot) + \mathbb{I}_{\{h < H\}} \sum_{s_+ \in S} p(s_+ \mid s, \cdot) \cdot v_k^i(s_+, h+1)$$

13:   update the beliefs and the Q-functions

$$\underline{\pi}_{k+1}^\ell(s, h) = \underline{\pi}_k^\ell(s, h) + \mathbb{I}_{\{s = s_{h,k}\}} \alpha_{c_k(s,h)} \cdot \left(\underline{a}_k^\ell(s, h) - \underline{\pi}_k^\ell(s, h)\right) \quad \forall \ell \neq m$$
$$Q_{k+1}^i(s, h, \cdot) = Q_k^i(s, h, \cdot) + \mathbb{I}_{\{s = s_{h,k}\}} \alpha_{c_k(s,h)} \left(\widehat{Q}_k^i(s, h, \cdot) - Q_k^i(s, h, \cdot)\right)$$

    for all $(s, h) \in S \times [H]$
14: **end for**

---

where $r_{h,k}^i$ denotes the reward received at stage $h$ at the $k$th repetition and we approximate the expected continuation payoff by looking one stage ahead, as in the classical Q-learning algorithm, and $c_k(s, h, a)$ is the number of times the pair $(s, a)$ gets realized at stage $h$ until the $k$th repetition. Algorithms 2 and 3 provide descriptions of the extensions, resp., for the model-based and model-free cases from the perspective of the typical agent $i \in \mathcal{I}^m$ from the typical team $m \in \mathcal{T}$.

*Remark* C.2. Algorithm 2 reduces to Algorithm 1 if $|S| = 1$ and $H = 1$.

## D  Large-scale Numerical Examples

In this section, we give a large-scale experiment that show the scalability of Team-FP in a networked game where only 2-hop neighbors of agents affect their payoff function. We simulated a three-team game with nine agents per team, resulting in a large joint action space of size $2^2 7$. After ten independent trials, we plotted the evolution of the Team-Nash Gap in Figure 7. Despite the problem's scale, the empirical averages of team actions converge to the Team-Nash equilibrium at a similar rate, even with sparse network interconnections, as shown in the top right of Figure 7.

**Algorithm 3** Model-free (Independent) Team-FP for MGs

---

1: **initialize:** $\{\underline{\pi}_0^\ell(\cdot)\}_{\ell \neq m}$, $\{a_{-1}^j(\cdot)\}_{j \in \mathcal{I}^m \setminus \{i\}}$, and $Q_0^i(\cdot)$ arbitrarily for the typical agent $i \in \mathcal{I}^m$
2: **for** each repetition $k = 0, 1, \dots$ **do**
3:     **for** each stage $h = 1, \dots, H$ **do**
4:         **require:** $\{\underline{\pi}_k^\ell(\cdot)\}_{\ell \neq m}$, $\{a_{k-1}^j(\cdot)\}_{j \in \mathcal{I}^m \setminus \{i\}}$, and $Q_k^i(\cdot)$
5:         observe current state $s$
6:         set $\bar{s} = (s, h)$
7:         play $a_k^i(\bar{s}) \sim \mathrm{br}_\tau(Q_k^i(\bar{s}, \cdot, a_{k-1}^{-i}(\bar{s}), \underline{\pi}_k^{-m}(\bar{s})))$ or $a_k^i(\bar{s}) = a_{k-1}^i(\bar{s})$ in a coordinated way (or independently) simultaneously with other agents
8:         observe $a_{h,k}^{-i} = a_k^{-i}(\bar{s})$
9:         receive $r_{h,k}^i = r^i(s, a_{h,k})$
10:    **end for**
11:    **require:** trajectory $\{s_{h,k}, a_{h,k}, r_{h,k}\}_{h=1}^H$
12:    set

$$v_k^i(s, h) = Q_k^i(s, h, \underline{a}_k^m(s, h), \underline{\pi}_k^{-m}(s, h))$$
$$\widehat{Q}_k^i(s, h, \cdot) = r_{h,k}^i + \mathbb{I}_{\{h < H\}} v_k^i(s_{h+1,k}, h+1)$$

13:    update the beliefs and the Q-functions

$$\underline{\pi}_{k+1}^\ell(s, h) = \underline{\pi}_k^\ell(s, h) + \mathbb{I}_{\{s = s_{h,k}\}} \alpha_{c_k(s,h)} \cdot \left(\underline{a}_k^\ell(s, h) - \underline{\pi}_k^\ell(s, h)\right) \quad \forall \ell \neq m$$
$$Q_{k+1}^i(s, h, a) = Q_k^i(s, h, a) + \mathbb{I}_{\{(s,a) = (s_{h,k}, a_{h,k})\}} \alpha_{c_k(s,h,a)} \left(\widehat{Q}_k^i(s, h, a) - Q_k^i(s, h, a)\right)$$

      for all $(s, h, a) \in S \times [H] \times A$
14: **end for**

---

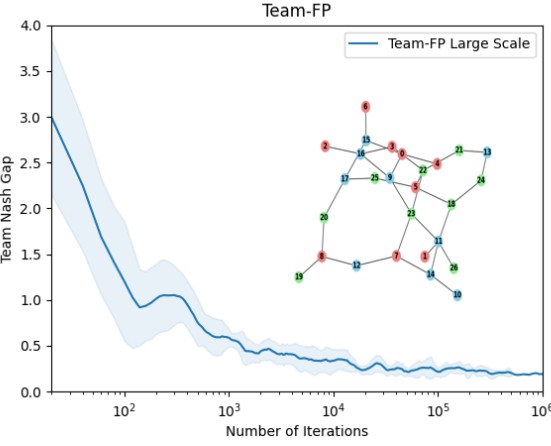

Figure 7: The evolution of Team Nash Gap in the large-scale example provided in the top right, showing that Team-FP dynamics reach near team-minimax equilibrium.

