# OpenReview forum: "Team-Fictitious Play for Reaching Team-Nash Equilibrium in Multi-team Games"
_NeurIPS.cc/2024/Conference — NeurIPS 2024 poster_

### Official Review · Reviewer_LmHf · 2024-06-18

**Soundness:** 1
**Presentation:** 1
**Contribution:** 1
**Rating:** 3
**Confidence:** 4

**Summary:**

This paper introduces a new variant of fictitious play where agents respond to the last actions of team members. The authors need to improve their academic writing skills largely. The expression between paragraphs and sentences lacks logic and consistency. Moreover, the paper does not list the challenges and gaps in current research and why this research can solve the recent issue. Furthermore, from my point of view, I did not see particular novelty or practical value in this research.

**Strengths:**

This paper introduces a new variant of fictitious play where agents respond to the last actions of team members.

**Weaknesses:**

The authors need to improve their academic writing skills largely. The expression between paragraphs and sentences lacks logic and consistency. Moreover, the paper does not list the challenges and gaps in current research and why this research can solve the recent issue. Furthermore, from my point of view, I did not see particular novelty or practical value in this research.

**Questions:**

I suggest the author reorganize the paper and improve the academic writing skills.

**Limitations:**

I suggest the author reorganize the paper and improve the academic writing skills.

---

> ### Author Rebuttal · Authors · 2024-08-07
>
> We disagree with the reviewer’s claim that expressions between paragraphs and sentences lack logic and consistency except possibly for very few instances. We would appreciate it if the reviewer could explicitly refer to any specific examples so that we can address them.
>
> The first two paragraphs of the introduction highlight the challenges and the gaps in the literature and how this paper can solve these issues.
>
> **What are the challenges and gaps in the current research?**
> * There are **no results** addressing whether Team-Nash equilibrium (TNE) can predict the outcome of interactions among multiple teams of autonomous decision-makers in non-cooperative environments despite the recent interest in multi-team games and the efficient computation of TNE.
> * Joint team responses can require the correlation of the responses among team members. Therefore, the widely studied fictitious play dynamics do not necessarily reach TNE even in two-team zero-sum games. There are also **no results** achieving such correlation without communication or ex-ante coordination for multi-team games.
>
> **Why can this research address this issue?**
> * We **present a slight adjustment** of the fictitious play dynamics in the direction of log-linear learning, another widely studied dynamics. This adjustment ensures that the dynamics involve simple behavioral rules consistent with the autonomous agents’ adaptation according to their self-interests, and team members can still learn to coordinate in the best team response against other teams without communication or ex-ante coordination.
> * We **prove the almost-sure convergence** to TNE in a broad class of multi-team games, including the two-team zero-sum and multi-agent zero-sum polymatrix games as special cases.
>
> **What are the practical novelty or value of this research?**
> * Our results **strengthen the predictive power of TNE** for multi-team games, **justify the recent interest** in efficient computation of TNE for multi-team games, and **provide a theoretical foundation** to design/regulate multi-team games under TNE.
> * We have exemplified the practical impact of the results on an **airport security scenario** in the global response. The example shows that designing airport security for the worst-case of the team-minimax equilibrium against autonomous attackers that do not coordinate is not too conservative since the attackers can learn to coordinate in the worst response by following simple behavioral rules, as in the Team-FP dynamics.
> * Due to the **scalability** of the Team-FP dynamics, teams of agents can also follow the Team-FP dynamics to learn the coordination in the best team response against other teams without the burden of communication and ex-ante coordination, especially in **large-scale systems with networked interconnections**. The global response includes such a large-scale numerical example.
>
> Regarding the ethics flag, we have **only used randomly generated games** in our paper without external data and provided our codes as supplementary material. There cannot be ethical issues regarding data representativeness and quality.

---

### Official Review · Reviewer_te9F · 2024-07-12

**Soundness:** 3
**Presentation:** 2
**Contribution:** 3
**Rating:** 5
**Confidence:** 3

**Summary:**

This paper addresses the complex problem of multi-team games by introducing a new variant of fictitious play called Team-FP. The method aims to enable teams of self-interested agents to reach Team-Nash Equilibrium (TNE) in multi-team games, with a particular focus on zero-sum potential team games (ZSPTGs). The authors present a rigorous convergence analysis and extend the Team-FP dynamics to multi-team Markov games. Extensive simulations are provided, comparing Team-FP with other algorithms to demonstrate its effectiveness and practical applicability.

**Strengths:**

The paper introduces Team-FP, a novel variant of fictitious play specifically designed for multi-team games. The approach incorporates inertia in action updates and agents’ responses to the last actions of team members, which enhances team coordination. This creative extension of classical fictitious play is valuable in advancing the understanding of team dynamics in multi-agent settings.

The paper is well-structured and organized, making it easy to follow the progression from the introduction of the problem to the presentation of the proposed methods.

**Weaknesses:**

Despite the novelty of the Team-FP approach, the modifications to classical fictitious play are not significantly groundbreaking. The methods, while creative, represent incremental improvements rather than major innovations in the field.

I am not familiar with the convergence proof; thus, I cannot and do not have the time to verify the proof. The authors put a lot of effort into the theoretical part. The experiment setting and results are simple. For a theoretical paper, it is better to prove the convergence through experimental results.

Thus, I think the paper is below the acceptance bar of NeurIPS.

**Questions:**

1. How can we get the Eq.4?

2. Can you provide experimental evidence to support your convergence proof?

---

> ### Author Rebuttal · Authors · 2024-08-07
>
> We thank the reviewer for their thorough and constructive comments. Firstly, we want to highlight that the equation (4) is inherited from polymatrix games. For example, the equation always holds for two-team games. As discussed in the introduction, the polymatrix structure (or network separable interactions) is essential for generalizing two-team zero-sum games to more than two teams.
>
> **[Novelty of the Team-FP]** We see this *slight* adjustment of the fictitious play dynamics in the direction of log-linear learning as a *strength*, justifying it as a simple behavioral rule consistent with the agents’ autonomous adaptations based on their self-interests. Notably, both dynamics have been studied extensively to model the interactions of human-like self-interested learners in the learning-in-games and behavioral-game-theory literature. Therefore, by showing the convergence of Team-FP dynamics to Team-NE, we provide **behavioral support for the predictive power of Team-NE** in multi-team environments (with uncoordinated team members), as exemplified in our **airport security scenario** in the global response. These convergence results also **justify the design of sophisticated equilibrium computation methods** to predict the outcome of such multi-team interactions. Furthermore, the simplicity of the dynamics ensures its **scalability for large-scale games** with networked interconnections, as discussed in the global response and its supplementary PDF.
>
> **[Experimental Support for Convergence]** Our extensive numerical simulations show convergence to Team-NE approximately as the Team-Nash Gap decays to near zero. The inherent exploration in softmax response induces the approximation error, and such occasional deviations from the greedy best response play an important role in escaping suboptimal team responses.

---

> > ### Comment · Reviewer_te9F · 2024-08-08
> > **thanks and raise score**
> >
> > Thank you for your response to my comments. I have read your rebuttal and am happy to raise the score.

---

> > > ### Author Response · Authors · 2024-08-12
> > > **Thank you!**
> > >
> > > We thank the reviewer for reviewing the rebuttal and raising the score.

---

### Official Review · Reviewer_Rftf · 2024-07-12

**Soundness:** 3
**Presentation:** 2
**Contribution:** 3
**Rating:** 6
**Confidence:** 2

**Summary:**

This paper introduces Team-Fictitious Play (Team-FP) dynamics as a novel approach for teams of self-interested agents to converge to Team-Nash equilibrium in multi-team games. The study focuses on games where multiple teams interact strategically, aiming to maximize their collective utilities. For this purpose, this paper (1) introduces Team-FP as a method for teams to converge to Team-Nash equilibrium in multi-team games; (2) extends the convergence analysis of Team-FP dynamics to multi-team Markov games; (3) demonstrates the effectiveness of Team-FP dynamics through theoretical analysis and empirical evaluations in various multi-team game scenarios.

**Strengths:**

1. The introduction of Team-FP represents a novel approach to address the challenge of teams reaching equilibrium in multi-team games.
2. The detailed numerical analysis and simulations conducted to evaluate the behavior of Team-FP dynamics in various multi-team game settings reflect the thoroughness and quality of the method.

**Weaknesses:**

1. The notations are confusing and it's hard to keep up. For example, 'agent index' and 'team index' both use lowercase letters.
2. Limited discussion on computational complexity. The paper could provide more insights into the computational complexity of implementing Team-FP dynamics in large-scale multi-team games. Discussing the scalability of the approach, potential bottlenecks, and computational efficiency considerations would be beneficial for understanding the practical feasibility of deploying Team-FP in complex settings.

**Questions:**

The authors provide some experiments, but it would be helpful to see more extensive experiments, including a larger-scale multi-team game to demonstrate the effectiveness of the algorithm in a more complex setting.

**Limitations:**

The authors have adequately addressed the limitations

---

> ### Author Rebuttal · Authors · 2024-08-07
>
> We thank the reviewer for their thorough reading of our paper and constructive comments. We appreciate the feedback regarding the notation. We have reserved uppercase letters to denote sets; e.g., $A^1$ denotes agent 1’s action set. For the explicit exposition, we underlined the parameters related to teams, e.g., $\underline{a}^1$ (and $\underline{\pi}^1$) denotes the joint team action (and team strategy) of team 1 while $a^1$ (and $\pi^1$) denotes agent 1’s local action (and local strategy).
>
> Since the Team-FP dynamics have linear updates, they are **scalable, especially for sparse networked interconnections**, as discussed in Remark 3.1 of the paper. For example, if at most 2-hop neighbors of the agents affect their reward (as in Example 2.1 of the paper), the computational complexity grows linearly with the number of teams and the maximum number of agents in a team. Furthermore, the complexity grows exponentially with the number of 2-hop neighbors of agents, where the base of the exponent is the number of local actions.
>
> To illustrate the scalability of the Team-FP dynamics, we have conducted experiments on a **large-scale game** as requested by the reviewer and described its details and the numerical results in the global response and its supplementary PDF. We also highlight that the global response includes the **practical application of an airport security scenario** modeled as a multi-team game.

---

> > ### Comment · Reviewer_Rftf · 2024-08-12
> >
> > Thanks for the clarifications and extra experimental results. I'd like to raise my score. However, it would be better if the authors could simplify their notations.

---

> > > ### Author Response · Authors · 2024-08-13
> > > **Thank you!**
> > >
> > > We thank the reviewer for reviewing the rebuttal and raising the score. As recommended, we will simplify the notation, e.g., use bold letters for team-related parameters, once the paper gets updated.

---

### Official Review · Reviewer_Jh6E · 2024-07-13

**Soundness:** 3
**Presentation:** 3
**Contribution:** 3
**Rating:** 6
**Confidence:** 3

**Summary:**

In this work, the authors introduce a novel variant of virtual play, referred to as **Team-Fictitious Play (Team-FP)**, aimed at assisting self-interested agents within teams to reach **Team Nash Equilibrium (TNE)** in multi-team games. The paper focuses on zero-sum potential team games (ZSPTGs), where teams interact pairwise, but the payoffs to team members are not necessarily identical. The main contributions include the introduction of inertia in action updates and responses to the last actions of team members, which are crucial for team coordination. The authors provide theoretical convergence guarantees and validate the efficacy of the approach through extensive simulations.

**Strengths:**

1. The introduction of Team-FP fills a gap in the multi-team game theory literature, particularly in the context of zero-sum potential team games.
2. The paper offers rigorous theoretical analysis and practical insights, including convergence proofs and error bounds.
3. Extensive simulations compare Team-FP with other algorithms, demonstrating its effectiveness and exploring the impact of various parameters on convergence speed.
4. The grammar and expression are accurate and professional.

**Weaknesses:**

- Adding more background or appendix sections on Nash equilibrium and related solution algorithms would greatly help the reader's understanding.
- The theoretical analysis section is quite technical; including more intuitive explanations and illustrations could be beneficial. For instance, adding a diagram in Section 3 to visually depict the workflow of Team-FP might help.
- A more detailed analysis of the parameters used in Team-FP and their sensitivity to performance could provide deeper insights.
- While the paper focuses on zero-sum potential team games, discussing how Team-FP could be adapted or extended to other types of multi-team games would be valuable.
- The numerical experiments section could benefit from more qualitative analysis.
- The discussion on practical applications is not sufficiently thorough; adding more experiments and analysis in specific application scenarios would be beneficial.
- Section 5 could include experimental comparisons with other multi-team learning algorithms.

**Questions:**

All my questions are written inside the weakness part.

**Limitations:**

The paper explicitly discusses its limitations.

---

> ### Author Rebuttal · Authors · 2024-08-07
>
> We appreciate the reviewer’s thorough review and constructive comments. We can gladly include further clarifications on the points raised, which will improve our paper’s accessibility. In the following, we address the reviewer’s questions:
>
> **[How can we visualize the learning dynamic and the solution approach?]** We have included Figure 1 to visually depict the Team-FP dynamics and the fundamental idea in our technical analysis in the PDF attached to the global response.
>
> **[How can we extend Team-FP to other classes of games?]** Similar to the FP dynamics, Team-FP dynamics are **game agnostic**. In other words, agents do not know whether the other teams have aligned or misaligned objectives. Furthermore, agents do not know whether other team members have identical or different yet aligned objectives. As discussed in the conclusion section, various essential classes of games have the fictitious play property. The generalization of our approach to such multi-team games is an interesting future research direction. In the illustrative examples section, we have discussed several other types of multi-team scenarios beyond zero-sum potential team games, such as **$2\times N$ general-sum game** and **potential-of-potentials game**. As observed in these examples, we expect the almost-sure convergence of Team-FP for the games where FP converges if teams were single agents since in the Team-FP dynamics, team members can learn to coordinate in the (evolving) best team response and can act as if they are a single decision-maker.
>
> **[How sensitive is the performance to the parameters used?]** In the illustrative examples section, we have numerically examined and discussed the parameters $\tau$ and $\delta$. Our results indicate that decreasing $\tau$ (i.e., reducing exploration) leads to a lower Team-Nash Gap at the steady state. However, exploration plays an essential role in escaping suboptimal team responses. Increasing $\delta$ (or reducing friction) can accelerate convergence. However, friction in the action updates is critical for reaching the optimal team response.
>
> **[How does Team-FP perform in practical applications?]** The reviewer raises a significant concern. In the rebuttal period, we have implemented a real-life application scenario to address this question. To this end, we generalized the widely studied **airport security scenario** to multi-team games. Attackers have aligned objectives due to their adversarial nature. Attackers are autonomous decision-makers with different objectives. Though their objectives can differ to a certain extent, they are aligned in maximizing the defender’s cost due to their adversarial nature. This makes our multi-team formulation an ideal model for such cases. We have described the game and presented the numerical results in the global response and its supplementary PDF. Our analysis shows that **defending our systems according to the worst-case as if the attackers are coordinated centrally is important since the attackers can learn such coordination based on simple behavioral rules consistent with their self-interests, as in the Team-FP dynamics.**
>
> **[How does Team-FP perform compared to other multi-team learning dynamics?]** The existing results on multi-team games are primarily computational. On the other hand, in this paper, we present simple behavioral rules (slight adjustments of the widely studied fictitious play dynamics) that can reach Team-NE to **strengthen the predictive power of Team-NE and justify the algorithms developed to compute Team-NE efficiently** for multi-team games, as exemplified in our airport security scenario. Furthermore, their simplicity makes the dynamics scalable for **large-scale problems**, as the global response discusses. To our knowledge, Team-FP is the first multi-team learning dynamic that does not rely on any (ex-ante) communication among team members. Notably, the Fictitious Team Play (FTP) algorithm, presented by [Farina et al., "*Ex ante* coordination and collusion in zero-sum multi-player extensive-form games," In NeurIPS, 2018], is related, and we have compared FTP and Team-FP dynamics in the global response and its supplementary PDF.

---

### Author Rebuttal · Authors · 2024-08-07

We thank the reviewers for their valuable time and constructive comments.

Based on Reviewer Jh6E’s request, we have illustrated the Team-FP dynamics and the fundamental idea of the proof in Figure 1 of the attached PDF. Furthermore, based on the other comments, we have conducted several new experiments despite the limited time and computational resources. We have also provided the results of these experiments in the PDF. In the following, we describe these experiments in more detail.

**[A Practical Application]** We can model an **airport security scenario** as a two-team game between defender and attacker teams. Note that airport security has been studied from a game-theoretical lens extensively, and the findings have been **deployed in real life**, e.g., see [An et al., “PROTECT - A deployed game theoretic system for strategic security allocation for the United States Coast Guard,” AI Magazine, 2013]. In our example, we consider a single security chief in the defender team and three different intruders in the attacker team, as illustrated on the left-hand side of Figure 2 of the PDF. The chief can defend a gate at the expense of some cost. Intruders autonomously decide whether to attack a specific gate or remain idle. The intruders receive positive (or negative) payoffs if they attack an undefended (or defended) gate. Correspondingly, the chief gets a positive (or negative) payoff if intruders attack defended (or undefended) gates. The security chief has $2^6=64$ actions, while each intruder has $6+1=7$ actions. We have conducted $50$ independent trials and presented the evolution of the average of the Team-Nash Gap along with the standard deviations as shaded areas on the right-hand side of Figure 2. From a higher level, this example shows that **team-minimax equilibrium can predict the outcome of airport security games for the likely scenario of different uncoordinated attackers. It also justifies the algorithms developed to compute team-minimax equilibrium efficiently.**

**[A Large-Scale Example]** We have simulated a three-team game with nine agents per team and $27$ agents. Note that this configuration has a huge joint action space of $2^{27}$ action profiles. We have conducted ten independent trials. We have plotted the evolution of the Team-Nash Gap in Figure 3 of the PDF. Despite the scale of the problem, our experiments show that the empirical averages of the team actions converge to the Team-Nash equilibrium **at a comparable rate when the networked interconnections are sparse**, as depicted in the top right of Figure 3.

**[A Comparison with Other Multi-team Learning Dynamics]** The existing learning dynamics for multi-team games are very limited as the literature mainly focuses on efficient computation of equilibrium rather than identifying whether equilibrium can **emerge** as an outcome of self-interested adaptations. **One exception** is the fictitious team play in [Farina et al., "*Ex ante* coordination and collusion in zero-sum multi-player extensive-form games," In NeurIPS, 2018]. In our setup, their fictitious team play dynamics (developed for extensive-form games) become the classical fictitious play dynamics where the entire team acts as a single decision-maker due to the ex-ante coordination among team members. Our paper examined this scenario in Figure 2(a) as a full coordination setup. In Figure 4 of the PDF, we have replotted this scenario to highlight the comparison.  Although the fictitious team play reaches equilibrium faster due to the ex-ante coordination, such coordination brings in additional burdens beyond time. In contrast, Team-FP dynamics are scalable without such coordination burden, as shown in the large-scale example above. Figure 2-(a) in the paper also shows the trade-off between coordination burden and learning speed by including the half-coordination scenario.

---

### Decision · Program_Chairs · 2024-09-25

**Decision:**

Accept (poster)

**Comment:**

The paper received 4 reviews, 3 of which were detailed and provided useful comments. These reviews appreciate the gap that this paper fills in the literature by introducing the team-fictitious play algorithm and proving its theoretical convergence properties in team zero-sum potential games. The theoretical results are substantiated by numerical experiments (partially in large games) some of which were provided during the rebuttal by the authors. There was an additional ethics review which raised some concerns which are standard to this type of research and which can be addressed by the inclusion of an appropriate discussion by the authors (as already done in their response to this review).

Based on the 3 quality reviews and the ethics review, and the authors-reviewers and reviewers-reviewers discussions, my recommendation is to accept this paper as a poster. The reason is that the paper presents a novel contribution to the learning in games literature and provides enough evidence to substantiate the importance of the proposed algorithm as a simple (and, thus, scalable) extension of fictitious play in team-games. Nevertheless, this remains a borderline decision since the scope and the contribution of the paper is potentially limited by the theoretical nature of the settings that it studies.

For the preparation of the camera-ready version, the reviewing team trusts that the authors will include the ethics discussion as promised in their response and the additional illustrations and experiments presented in their attached document. Also, the reviewing team encourages the author to acknowledge the other important comments and limitations raised in the discussions with reviewers.